# Predicting areas important for ecological connectivity throughout Canada

Richard Pither[1]*, Paul O'Brien[2], Angela Brennan[3,4], Kristen Hirsh-Pearson[5], Jeff Bowman[2,6]*

1 National Wildlife Research Centre, Environment and Climate Change Canada, Ottawa, Canada, 2 Ontario Ministry of Natural Resources and Forestry, Peterborough, Canada, 3 Interdisciplinary Biodiversity Solutions Program, University of British Columbia, Vancouver, Canada, 4 Institute for Resources, Environment and Sustainability, University of British Columbia, Vancouver, Canada, 5 Conservation Solutions Lab, University of Northern British Columbia, Prince George, Canada, 6 Trent University, Peterborough, Canada

* Richard.pither@ec.gc.ca (RP); jeff.bowman@ontario.ca (JB)

**Data Availability Statement:** Our input movement cost surface layer, output current density map, and code used for the analysis are available in the OSF database (https://osf.io/z2qs3/); DOI 10.17605/OSF.IO/Z2QS3.

## Abstract

Governments around the world have acknowledged that urgent action is needed to conserve and restore ecological connectivity to help reverse the decline of biodiversity. In this study we tested the hypothesis that functional connectivity for multiple species can be estimated across Canada using a single, upstream connectivity model. We developed a movement cost layer with cost values assigned using expert opinion to anthropogenic land cover features and natural features based on their known and assumed effects on the movement of terrestrial, non-volant fauna. We used Circuitscape to conduct an omnidirectional connectivity analysis for terrestrial landscapes, in which the potential contribution of all landscape elements to connectivity were considered and where source and destination nodes were independent of land tenure. Our resulting map of mean current density provided a seamless estimate of movement probability at a 300 m resolution across Canada. We tested predictions in our map using a variety of independently collected wildlife data. We found that GPS data for individual caribou, wolves, moose, and elk that traveled longer distances in western Canada were all significantly correlated with areas of high current densities. The frequency of moose roadkill in New Brunswick was also positively associated with current density, but our map was not able to predict areas of high road mortality for herpetofauna in southern Ontario. The results demonstrate that an upstream modelling approach can be used to characterize functional connectivity for multiple species across a large study area. Our national connectivity map can help governments in Canada prioritize land management decisions to conserve and restore connectivity at both national and regional scales.

## Introduction

The loss and degradation of ecological connectivity is considered an important driver of global biodiversity loss [1–3]. This has been formally acknowledged by governments around the world, including through the United Nations (UN) General Assembly, which issued Resolution 75/271 in 2021 that stressed the need for cooperation "on the enhancement of

**Funding:** This project was undertaken with the financial support of the Government of Canada through the federal department of Environment and Climate Change Canada and the Government of Ontario through the Ministry of Northern Development, Mines, Natural Resources and Forestry.

**Competing interests:** The authors have declared that no competing interests exist.

connectivity between ecosystems and cooperation in order to maintain healthy and intact ecosystems and habitats, which are needed to conserve biodiversity. . .". The first goal in the recently adopted UN Convention on Biological Diversity (CBD) Post-2020 Global Biodiversity Framework states by 2050, the ". . . integrity, connectivity and resilience off all ecosystems are maintained, enhanced, or restored,. . ." [4]. Countries will now have to accelerate efforts to identify and conserve areas important for connectivity, and federal governments would benefit from country-wide data to help objectively prioritize areas for financial and logistical support.

Prioritizing areas over broad extents, however, can be challenging given that connectivity, or the "degree to which the landscape facilitates or impedes movement among resource patches" [5] is a species dependent feature. Large study areas will have many species with a variety of movement traits and resource needs, such that areas important for connectivity for one species may not be important for others [6]. Despite this challenge, the need to halt the decline of biodiversity is urgent, and therefore strategic approaches are required to capture the connectivity needs for multiple species [7, 8]. In general, those needs can either be captured "downstream", by producing species-specific connectivity models for several species and then combining the results into a single connectivity map, or "upstream" by producing a single connectivity model that is assumed to capture the needs of multiple species [9]. The downstream approach has the potential to produce more biologically realistic models that directly estimate functional connectivity (i.e., based on data for animal movement through landscapes), but they require a lot of species-specific data, which is often not available or is difficult and expensive to obtain [7, 10].

In contrast, upstream approaches attempt to model functional connectivity by incorporating the movement behaviours of multiple species across generalized environmental attributes and land cover types. The movement patterns of many species, for example, are known to be negatively affected by human-modified land cover features such as cities and roads [11–14]. Indeed, a global study found that the movement of 57 mammal species was reduced by up to one half in areas with a high human footprint [15]. Based on such evidence, studies have used some form of a human footprint (or its inverse, naturalness) for their movement cost (or resistance) layer to assess and map connectivity at local [16], national [17–19] and even continental scales [20]. Given the generality of this method, the movement cost layer is typically constructed using more expert opinion than would be the case for single-species analyses, which are largely informed by animal movement data or genetics [21]. Even so, Koen et al. [22] found that an upstream model was able to predict areas of functional connectivity for multiple species in their study area in eastern Ontario, Canada. Consequently, this modelling approach might be suitable for characterizing multi-species connectivity across a large, diverse region with varying data availability and species diversity. However, to date no broad scale applications of this method have evaluated how well the models predict functional connectivity for multiple taxa using independent data.

Canada is a member state of the United Nations and a party to the CBD and has indicated its support for global targets. To date, the only national-scale analysis of connectivity across Canada was restricted to the structural connectivity (i.e., based on the spatial arrangement of habitat, irrespective of how animals are moving) of the country's forested regions using a resistance layer with a low thematic resolution of forest vs non-forest [23]. Although that study was able to use a resistance surface with a fine spatial grain (25 x 25 m), it required dividing Canada into thousands of small tiles (25 x 25 km) due to computational limitations. Koen et al. [24] have since determined that the accuracy of current density maps declines with decreasing tile size. At the continental-scale, Carroll et al. [25] identified areas important for connectivity between current climate zones and their future analogs under various climate change projections in North America, but at a coarse spatial resolution (5 x 5 km). Barnett and Belote [20]

used the same 5-km resolution to model an aspirational network of connected protected areas across North America. More recently, Brennan et al. [26] assessed the connectivity of terrestrial protected areas for the entire world, but again at a relatively coarse spatial resolution and only among protected areas larger than 35 km$^2$.

Mapping connectivity among existing protected areas is important, but in many countries protected areas represent only a small portion of the landscape [27]. In fact, only 12.5% of Canada's terrestrial areas are currently protected [28], less than half of the 30% target in the Post-2020 Global Biodiversity Framework [4, 29]. In addition, research has shown that when connectivity is analysed using circuit theory and least cost models, the resulting connectivity patterns strongly depend on the placement of 'nodes' marking the source and destination of animal movement (e.g., protected areas or core habitats); that is, the same landscape will produce widely differing outputs depending on node placement [22]. As a result, specifying nodes can obscure connectivity between unknown (i.e., unspecified) sources and destinations of movement. This may be a critical omission, particularly for studies looking to examine connectivity across large or national scales for multiple taxa, where all sources and destinations of movement cannot be known. For these applications, an omnidirectional approach with nodes placed in buffers outside of the study area may be the best method to produce a seamless, multi-species current density map that is not dependent on specific node placement.

In this study we tested the hypothesis that functional connectivity for multiple species can be modelled across Canada using expert opinion and an upstream approach [9]. For our test of functional connectivity, we predicted that current density from circuit theory would be positively related to independent estimates of animal movement for multiple species from different regions of the country. We used two recently developed national land cover layers (the Canadian Human Footprint and a road layer), along with layers for natural features known to affect the movement of terrestrial, non-volant fauna to construct a movement cost layer. We used Circuitscape to conduct an omnidirectional connectivity analysis for terrestrial landscapes, in which the potential contribution of all landscape elements to connectivity were considered and where source and destination nodes were independent of land tenure.

## Methods

### Movement cost surface

Analyzing connectivity requires an input movement cost (also called resistance) layer representing an estimation of the degree to which different land cover types affect the movement of individual animals or propagules [30, 31]. A land cover that physically slows movement, is avoided, or imparts a physiological cost is assigned a high cost (resistance) value; land cover types that are more likely to be used and crossed successfully are assigned a low cost. Cost surfaces can be constructed using expert opinion or empirical data [31]. Much like Koen et al. [22], we modelled connectivity for terrestrial, non-volant fauna that use and move across natural land cover more successfully than anthropogenic land cover types by assigning higher costs to landscape elements with higher degrees of anthropogenic disturbance. We ranked and assigned cost values to the anthropogenic layers using our own knowledge but in consultation with experts (see Supporting Information) and consistent with previously published studies [6, 17–20, 22, 26, 32–35].

We constructed our cost surface by combining land cover layers from 23 sources. Eight of those layers were from the Canadian Human Footprint (CHF) [36] and included built environments, nighttime lights, croplands, pasturelands, dams and reservoirs, mining, oil and gas, and forestry areas. We used a recently developed national road layer for Canada [37] that includes resource-access roads, along with a national railway layer. For the buffer area outside

of Canada, we used six layers from the Global Human Footprint [38] (GHF): built environments, nighttime lights, croplands, pasturelands, railways, and roads. In addition to human modified land cover features, we included natural features considered to affect the movement of terrestrial, non-volant fauna, namely elevation and slope, glaciers, permanent sea ice, as well as large lakes and rivers [6, 39–42] for within both Canada and the buffer areas in the United States of America (hereafter, U.S.). Vector layers (e.g., roads, railways, rivers) were rasterized and all input layers were resampled to a resolution of 300 x 300 m, where necessary, to match the resolution of layers from the CHF. Further details, including data sources, are available in the S1 Table.

We were interested in using current density maps from circuit theory as our main model output to predict functional connectivity for multiple taxa. Current density is proportional to the probability of use during a random walk [29]. Our model produced an estimate of current density between all possible pairs of nodes while accounting for movement costs. Current density maps have low sensitivity to the absolute value of the costs assigned to land cover types so long as the rank of the types is maintained [43]. For example, if crossing a freeway is more costly than crossing a field, current density will be similar among resulting maps regardless of the absolute costs applied to the freeway and field, provided that the freeway is always assigned a higher cost. However, the range of cost values (e.g., 1 to 3 versus 1 to 1000) does appear to have a small effect on the pattern of current densities [43, 44]. To determine whether these relationships remained true for our large, real-world study area, we tested the sensitivity of current density outputs with ten cost scenarios (S2 Table) using two regions of Canada (southern British Columbia–B.C., and the Maritime provinces of New Brunswick, Prince Edward Island, and Nova Scotia). For each region, we used Spearman rank correlations to compare the generated current density values for the same 1000 randomly selected cells between pairs of scenarios. We then selected the cost scenario that was most strongly correlated with all the others for use in the subsequent analyses.

We assigned a high movement cost to human-dominated land cover types (e.g., cities, major highways), a medium-high cost to human-impacted features (e.g., agricultural lands, minor highways), and a medium-low cost to more permeable human-modified land cover types (e.g., pasture lands, resource roads, harvested forests). High movement costs were also assigned to natural features known to inhibit the movement of terrestrial, non-volant fauna based on data from published studies and consultations with experts (S1 Table). These included areas with high elevations (> 2300 m [40, 41, 45]), steep slopes (> 30 degrees [39, 42, 45–47]), large lakes (> = 10 ha) and rivers with large flow (> 28 m$^3$/sec [6, 34, 48]). Conversely, because some mammal species are known to travel across permanent sea ice among islands in the Canadian Arctic Archipelago, we assigned a medium-low cost to that layer [49]. A single Canada-wide, cost surface raster was constructed by using the highest cost for a given pixel from among the 23 layers. Pixels that had no data (i.e., did not have any of the 23 land cover types) were assigned the low movement cost.

## Circuit theory

Over the last decade, circuit theory has become one of the more commonly used approaches to assess and map connectivity [50–52]. Most applications of circuit theory have tended to connect source and destination locations (i.e., 'nodes', as previously mentioned), to map all probable paths of movement between those nodes, measured as electrical current density (hereafter referred to as current density). However, because node placement can affect the resulting patterns of connectivity [22, 53], omnidirectional connectivity methods are being explored more frequently when the objective is to predict connectivity across landscapes in general as

opposed to among specific locations [54]. This can be done using an omnidirectional connectivity approach, which is particularly valuable when the source and destination of all movement is unknown and is highly relevant for generalized land use planning activities [6, 22, 55, 56]. The three main omnidirectional methods (wall-to-wall, point-based, Omniscape) have been described by Phillips et al. [54], who found that all three produced very similar current density patterns, while the wall-to-wall method was the fastest for computer processing.

We used the wall-to-wall omnidirectional, Julia implementation, advanced mode of Circuitscape [57] to run the circuit theory models on each tile following methods outlined by Phillips et al. [54]. The wall-to-wall method uses thin, one-pixel wide source and ground strips placed along opposite sides of a tile. Circuitscape was run in each of the four cardinal directions (east to west, west to east, north to south, and south to north) for a total of four runs per tile. The four directional runs were then combined by taking the mean of all four to produce an omnidirectional current density map for a given tile.

To increase computational efficiency, we divided Canada into a series of large tiles that were analysed individually and then stitched together to produce a national current density map. Knowing that we would likely be using tiles of different sizes because of the shape of the country, and knowing that tile size can affect current density estimates [24], we conducted a sensitivity analysis comparing current density across tiles of varying sizes. We tested ten tiles of sizes ranging from 25 x 25 to 1500 x 1500 cells. To control for effects related to the composition and spatial distribution of cost values, we simulated simplistic landscapes in which all tiles were split in two, with 50% of the cells assigned a high cost and 50% assigned a low cost. We then calculated and compared the minimum, maximum, mean and standard deviation of the current densities for each tile size to identify the size above which variation in current density estimates was minimal. We used the results from the sensitivity analyses to divide our study area into tile sizes above that threshold but that would still allow reasonably efficient run times with our computing resources. Tiles were centred on provinces and territories where possible, but some jurisdictions required multiple tiles because of their large sizes (S1 Fig). Tiles included a buffer on each side to reduce the effects of node placement and artificial map boundaries on current density patterns [22, 53]. Following Koen et al. [22], we used a buffer width equivalent to 20% of the average length of the sides of the tile of interest. This included buffers into the U.S. and surrounding oceans.

All current density maps were stitched together into a single map after their buffers and overlapping areas were removed. For the few cases where adjacent tiles overlapped within a single province or territory (e.g., Ontario; S1 Fig), we computed the mean of the overlapping areas. In two regions, however, taking the mean of the overlapping tiles still resulted in noticeable seams. To address those anomalies, we reran Circuitscape on additional tiles centred over those seams to produce new current density maps. We then computed the mean of all maps (the new maps and the originals) to create a single, seamless, national current density map. Our resulting model output was a seamless estimate of the probability of movement during a random walk for a set of terrestrial species that use natural cover across Canada.

### Independent movement data

We used independent wildlife datasets from across Canada to test for a relationship between current density and animal movement. We sought datasets for animal species that we considered would use natural cover and be limited in anthropogenic habitats. Consequently, we did not include data for peri-urban species (e.g., raccoons, skunks, coyotes) [58–60]. We were also interested in selecting species from a range of environments and geographical locations. We used GPS-collar data accessed through Movebank.org for moose (*Alces alces;* $N_{id}$ = 19; $N_{obs}$ =

55,181) [61, 62], grey wolf (*Canis lupus;* $N_{id}$ = 68; $N_{obs}$ = 174,441) [63], Rocky Mountain elk (*Cervus canadensis;* $N_{id}$ = 175; $N_{obs}$ = 1,585,428) [64, 65], and mountain caribou (*Rangifer tarandus;* $N_{id}$ = 186; $N_{obs}$ = 245,449) [66] located in British Columbia and Alberta, as well as road-kill data for herpetofauna in southern Ontario and moose in New Brunswick (see S2 Fig for general locations).

For the GPS-collar data from western Canada, we compared current density values extracted from observed locations (i.e., locations used by collared individuals) to current density values extracted from randomly selected available, but unused locations (hereafter referred to as available locations). For the observed locations, we randomly selected 10 percent of each individual's GPS data, as this is a common testing ratio used in habitat model validations [67] and can reduce the effects of autocorrelation. To obtain equal numbers of available locations we used the adehabitatHR package [68] in the R statistical computing environment [69] version 3.6.3 to first estimate each individual's home range using a 95% utilization distribution (UD) from the individual's full GPS dataset. We used the 95% UD to exclude areas with the lowest probability of use and thus strengthen the contrast between used and available locations. Next, we calculated each individual's maximum displacement distance, as the greatest straight-line distance (km) from any observed location to the earliest observed location. We then buffered each individual's UD by their maximum displacement distance and randomly selected available locations within this buffered area. We used boxplots to compare overall current density values between observed and available locations. We also examined current density between observed and available locations using species-specific linear mixed effect models with current density as the response variable, case (i.e., observed location = 1; available location = 0) as the categorical predictor and individual as the random effect, to account for possible correlation in current density values among each individual's observations. Linear mixed effects models were conducted in R using the lmer package [70].

After initial examination of these results, we questioned whether the difference in current density between observed and available locations was influenced by displacement distance; specifically, whether those current density differences became greater with increasing maximum displacement distance. We hypothesized that individuals who travel farther have to contend with more anthropogenic factors and barriers, thus traveling through more areas of concentrated current density (i.e., pinch points). To test this hypothesis post hoc, we used the same species-specific models previously described, with the addition of an interaction between case and individual maximum displacement distance (as described above). To clarify our discussion of these results, we binned individuals into two groups: short-distance movers identified as those having maximum displacement distances in the lower quartile bin (values in the 0–0.25 quantile range), and long-distance movers as those having maximum displacement distances in the upper quartile bin (values in the 0.75–1.0 quantile range).

The herpetofauna roadkill data from Ontario contained 4496 locations of 17 species killed along roadways in southern Ontario (south of the French River) between 2010 and 2019. We used t-tests and Cohen's Effect Size d to compare the mean current densities of the roadkill locations to densities found within an equal number of random locations along the same roads. For each location, we used the mean of the current densities of the focal pixel and within a 300 m buffer around the point data to capture current density adjacent to the roads.

We also assessed roadkill of moose in New Brunswick between 2001 and 2020, but only from highways with no wildlife fencing or passages to avoid biasing the analysis. Even so, moose roadkills were numerous, often with several observations in a given location. Moose are very large and not likely to be missed when dead on the side of the road and so we assumed that we had an accurate count of moose roadkills. Consequently, we divided roads into 900 m

sections and used a Spearman rank correlation to compare the frequency of moose roadkills to the mean current densities within sections and immediately adjacent to the roads.

## Results

Our sensitivity analyses found that the current densities generated using ten cost scenarios were all highly correlated (S3A Fig). Some of the minor variation in correlation values among pairs of cost scenarios can be attributed to differences in the range of cost values (S3B Fig; ranges in S2 Table). Some pairs of scenarios with large differences in the range of costs employed had lower correlations. In addition, scenarios with broader ranges of costs values resulted in higher maximum current densities (S3C Fig).

We constructed a movement cost map (Fig 1) using the cost values associated with the scenario that was the most strongly correlated with all other scenarios (scenario C9, S2 Table; S3A & S3D Fig) and that included a low cost of one because null values cannot be used in Circuitscape and to be consistent with previous studies [48, 71]. The cost map illustrates that much of the country still has land cover with low movement costs, although the presence of intense human activities (e.g., cities, croplands) can be seen clearly in southern Canada.

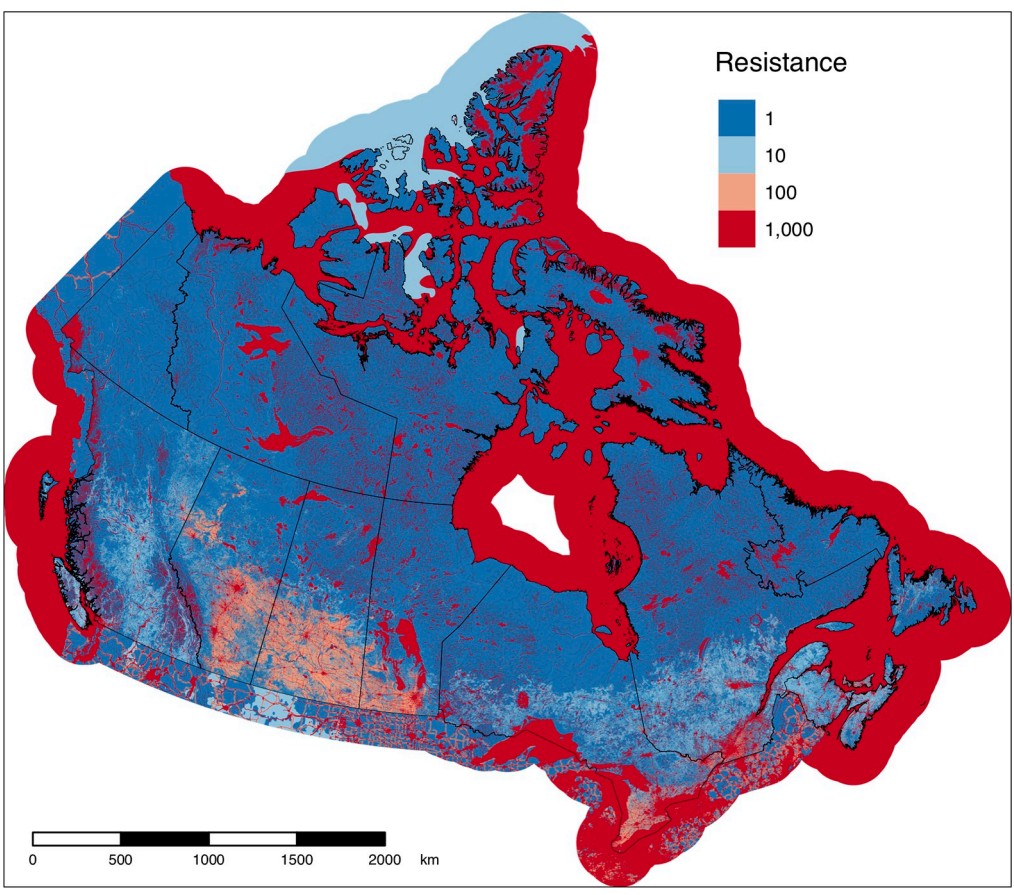

**Fig 1. Movement cost layer for Canada.** Four movement costs, reflecting the increasing cost of (or resistance to) movement of terrestrial, non-volant fauna across various land cover types, were used to construct the layer. Areas with intense human use, along with large lakes and rivers and steep slopes, have higher movement costs than natural areas. Contains information licensed under the Open Government Licence–Canada.

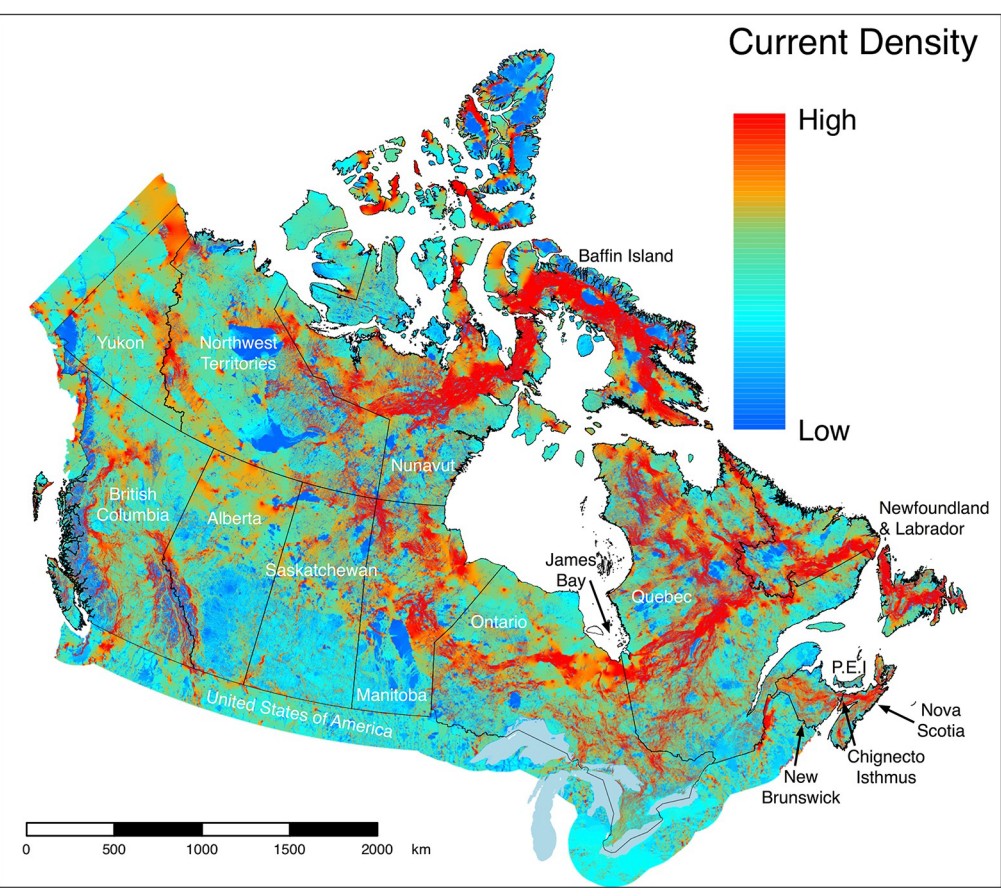

**Fig 2. A current density map identifying areas important for connectivity throughout Canada.** Areas with high current density (amperes) predict a high probability of movement by fauna that use natural cover and represent 'pinch points', which, if lost, could potentially limit connectivity. The omnidirectional, wall-to-wall method of Circuitscape [30] was used to estimate current density based on a movement cost surface composed of anthropogenic land cover features and natural features known to affect the movement of terrestrial, non-volant fauna. Contains information licensed under the Open Government Licence–Canada.

The sensitivity analysis of the tile sizes revealed that variations in current density estimates became negligible above sizes of 150,000 km$^2$ (1.7 million pixels; S4 Fig). Computational limitations meant that the largest tile we could efficiently analyse was 3,600,000 km$^2$ (40 million pixels). This required us to divide Canada into 17 rectangular tiles for the initial analysis. Addressing the anomalies at two of the seams required analysis of another two tiles, for a total of 19 tiles (S1 Fig).

The raw current density map (Fig 2) revealed heterogenous patterns of current density across the country, strongly influenced by Canada's unique geography, geometry, natural barriers, and human development. Large corridors of high current density were evident in many regions, especially those proximate to bodies of water or other large barriers. For example, large areas of high current density occurred just south of James Bay, across Baffin Island, and across the Chignecto Isthmus between New Brunswick and Nova Scotia.

In southern Canada, where the vast majority of species at risk reside [72] (Fig 3), patterns of current density are difficult to see at the scale of the national map. However, when viewed at finer resolutions (Fig 4A–4C), areas important for connectivity can be seen in regions considered priorities for biodiversity even within landscapes dominated by human activity. In

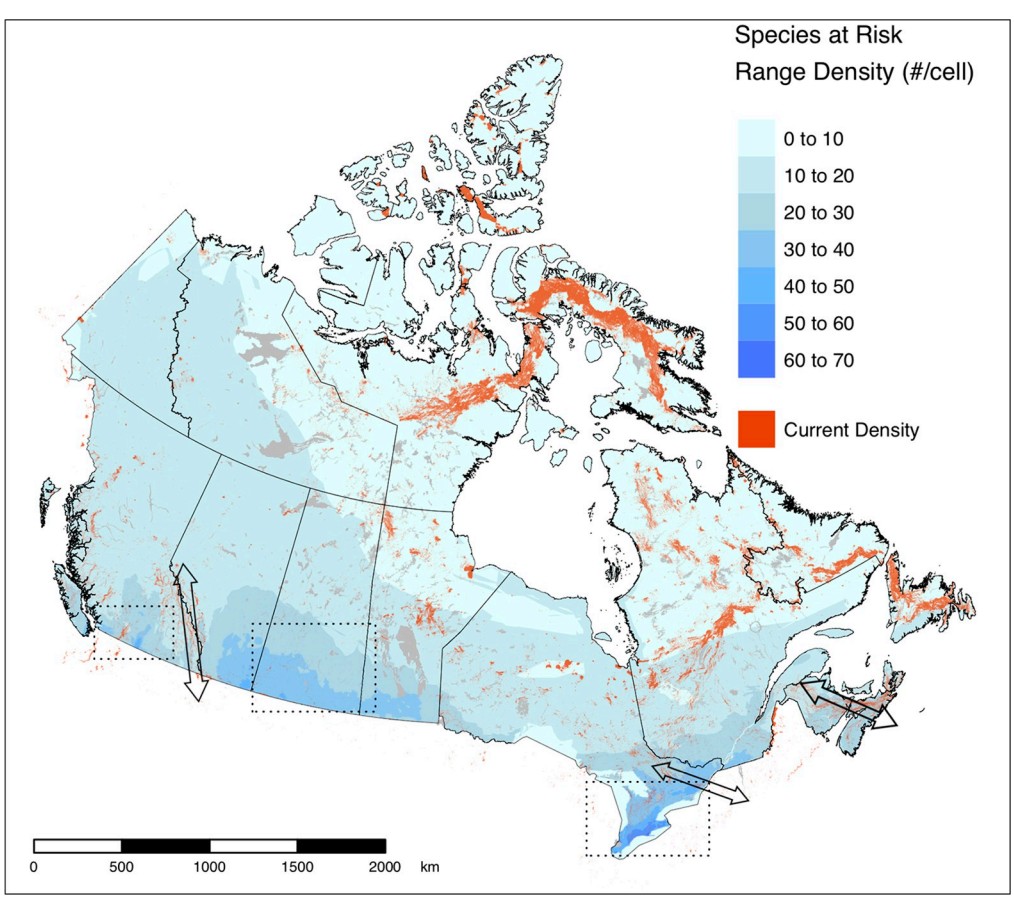

**Fig 3. Areas with high current densities over the density of ranges for species at risk in Canada [68].** The 95th percentile of current densities are displayed in orange. The three dashed boxes indicate the locations of the vignettes (Fig 4A–4C) and the black arrows provide examples of regions with existing connectivity conservation initiatives (Yellowstone to Yukon, Algonquin to Adirondacks, and the Chignetco Isthmus, from left to right). Contains information licensed under the Open Government Licence–Canada.

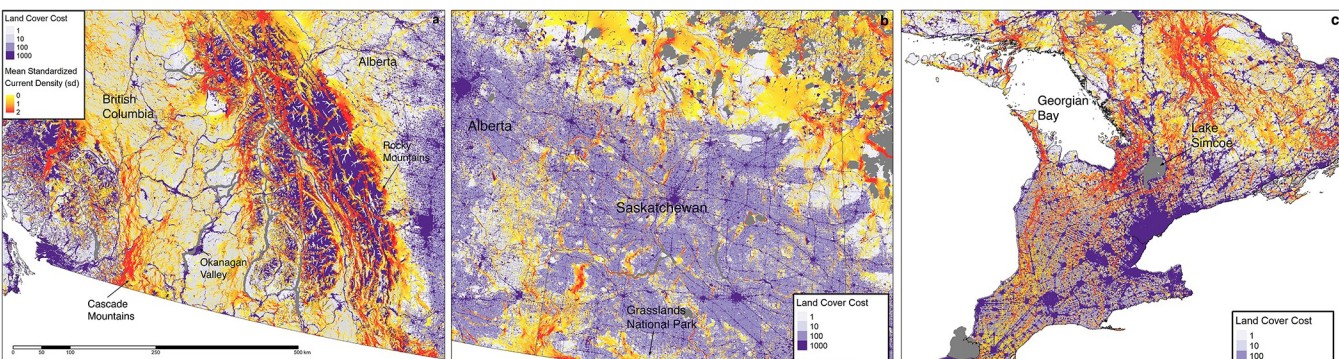

**Fig 4. Vignettes of three priority regions for species at risk and biodiversity in Canada.** Vignettes display a) southern British Columbia, b) southern Saskatchewan, and c) southern Ontario. Only current densities above the mean are displayed over the movement cost layer. Contains information licensed under the Open Government Licence–Canada.

southern British Columbia (Fig 4A) where populated areas are found in the valleys such as the Okanagan, options for movement remain in the upland resource management areas and higher mountain ranges, as reflected by diffuse patterns of current flow. Extensive agriculture throughout the prairies (Fig 4B), in contrast, leads to concentrations of high current densities following land less suitable to crops, including pasture lands used for cattle grazing. In southern Ontario (Fig 4C), Lake Simcoe is a prominent feature that leads to a north-south corridor of high current density along the shore of Georgian Bay.

## Independent movement data

In our initial species-specific linear mixed effects models of current density, we found the current densities among observed locations to be greater on average than current densities among available locations for grey wolf (0.81, 95% CI = 0.77, 0.86), moose (0.7, 95% CI = 0.65, 0.77, and mountain caribou (0.2, 95% CI = 0.20, 0.22, but not elk (-0.06, 95% CI = -0.08, -0.05). Similar findings are illustrated in the simple boxplots comparing current densities (Fig 5). However, after including an interaction with displacement distance, we found that for all of the species' long-distance movers, current density values were significantly higher in the observed locations compared to the available locations. Also, as we suspected, this effect increased with increasing distance moved (Fig 6 and S3 Table; and see S5 Fig for histograms of individual maximum displacement distances and their respective binning as short or long-distance movers). Herpetofauna roadkill were not found more often in locations with higher current densities, compared to random locations along the same roads (Fig 5; n = 4496, t = -2.02, P = 0.98, d = 0.04). There was a positive, albeit weak correlation between the frequency of moose roadkills in New Brunswick and current densities in the immediate area (Fig 7; n = 1216, ρ = 0.13, P < 0.001).

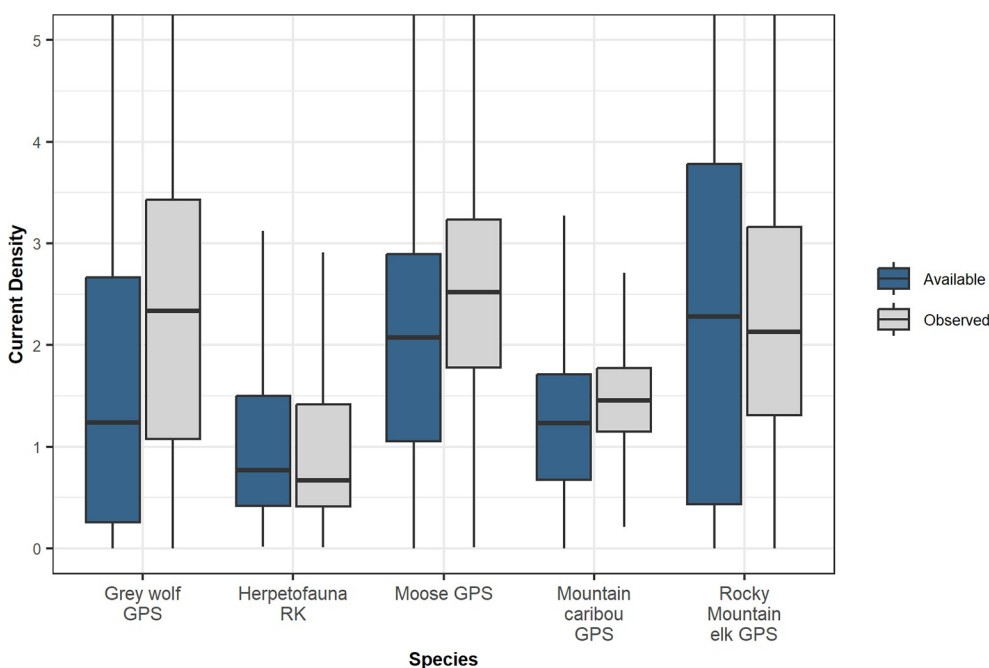

**Fig 5. Comparison of current densities at observed GPS locations versus available, but unused locations.** Boxplots depict median current density (thick horizontal line) and the central 50% of current density values (boxes) for observed wildlife locations and available, but unused locations. Extreme values were omitted from the figure for visualization purposes. GPS = data collected from GPS collars; RK = roadkill data.

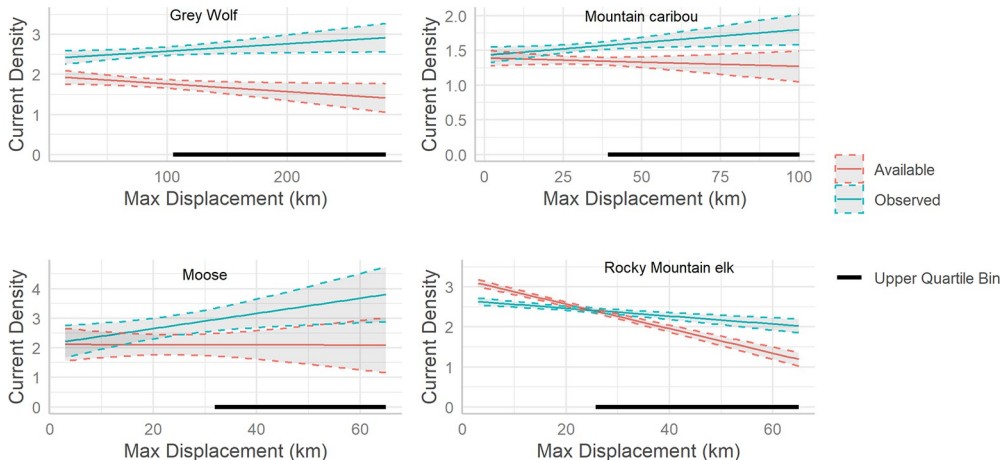

**Fig 6. Predicted interactions from the species-specific linear mixed effects models of current density.** Solid lines depict predicted mean current density for observed and available locations across the range of individual maximum displacement distances. Predictions were generated from a linear mixed effects model of current density with individual as a random effect and a fixed effect interaction between case (observed or available location) and individual maximum displacement distance. Dotted lines depict 95% confidence intervals; solid black lines indicate the max displacement distances in the upper quartile bin (i.e., the long-distance movers). Preference for areas with higher current density is suggested when current density values at observed locations are greater than at available locations.

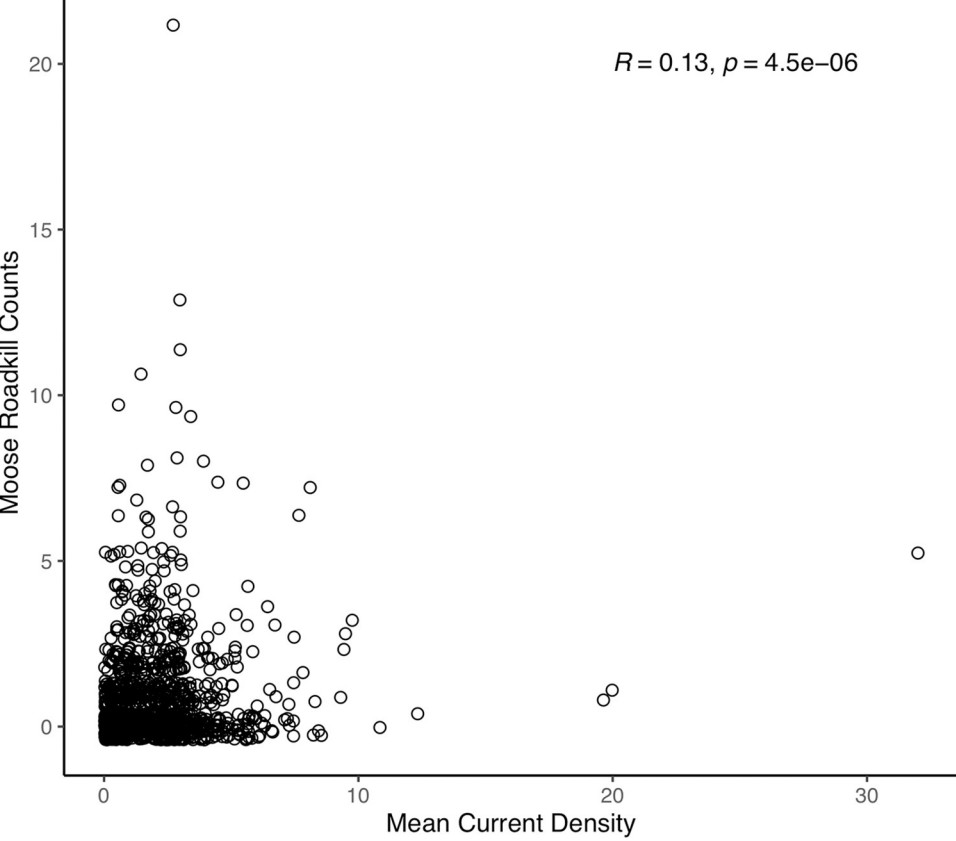

**Fig 7. Relationship between number of moose killed and mean current densities along roads in New Brunswick, Canada.** There was a small, positive correlation between the frequency of roadkills on and the current densities within 900x900m windows along roads.

## Discussion

Our findings suggest that an upstream modelling approach can be used to model landscape connectivity for multiple species across a large study area. As with other studies [6, 17–20, 26, 32, 34, 73], we used a version of a human footprint to create an input movement cost layer for our upstream model. This method is based on evidence that many species are more likely to use and move successfully through natural land cover types and conversely, avoid areas with more anthropogenic impact [11, 12, 74, 75]. Use of an upstream, generic species approach [9] and a human footprint could be considered a model of structural connectivity. However, we attempted to include elements of functional connectivity by ranking our movement costs based on published studies of species' biology and employing a random walk algorithm to model movement. Furthermore, the predictions of our model were supported by independent movement data for some species. Indeed, we consider that there is a gradient between the concepts of structural and functional connectivity, and that our study has aspects of both types. Our approach complements typical conservation policies and initiatives that often focus on protecting natural landscape features. Upstream connectivity approaches, therefore, offer promise for capturing the connectivity needs of multiple species to help address urgent conservation needs across large, geographically diverse areas in a timely fashion.

Upstream connectivity modelling approaches have been used for a few nation-wide analyses, primarily to identify ecological corridors among protected areas, including in China [19] and the U.S. [17]. In contrast to those studies, we assessed omnidirectional connectivity to avoid biases in current density patterns due to node placement [22] and because it is not possible to know all sources and destinations for multiple species in a country as large and diverse as Canada. In addition, only Riggio and Carro [18] have validated the accuracy of the predicted wildlife corridors at a national level, in their case using community knowledge in Tanzania. Indeed, it is surprising how few studies have attempted to validate the results from their connectivity models, including single species studies [8, 9, 52].

We tested predictions in our current density map using a variety of independently collected wildlife data from across the country and found support for its use as a map of functional connectivity for multiple taxa. We found that for individual mountain caribou, grey wolves, moose, and Rocky Mountain elk who traveled across larger areas, there was evidence of a stronger preference for areas with higher current density values compared to individuals who covered smaller distances. We also found that roadkill moose data in eastern Canada were positively correlated with areas of high current density.

With the Rocky Mountain elk, however, we found no evidence of preference for higher current density values among the short distance movers. The data used in our analysis originate from an elk population located in and around Banff National Park and the Ya Ha Tinda Ranch [76], which is an important elk winter range and a federally operated working horse ranch with human presence and modifications. Many of the elk that we classified as short distance movers were nonmigratory, year-round residents of the Ya Ha Tinda Ranch, and were previously found to have lower risk of predation by occupying the ranch compared to those individuals who migrated [63]. The human modified areas of the ranch are largely covered by lower current density values, which are then immediately surrounded by concentrations of higher current flow. The elk we classified as short distance movers occupied both areas, resulting in no apparent evidence of preference for high current densities.

Our current density map was not able to predict areas of high road mortality for herpetofauna in southern Ontario. We suspect that the spatial resolution of our map (300 m) is not fine enough for such small taxa, especially since a previous study using 100 m resolution data for eastern Ontario found that herpetofauna roadkill were located more often in areas with

high current density [22]. And although we found a positive relationship between the frequency of moose roadkills and current densities proximate to the roads in New Brunswick, current density explained only a small amount of the variation in roadkill. This suggests that there are other factors having a stronger influence on roadkill locations and so should be investigated in advance of locating mitigation measures. Other studies have found, for example, that hotspots for roadkill are related to the proximity of specific landcover types [77], such as wetlands for amphibians [78]. Canal et al. [79] found that the risk of roadkill depended on very fine-scale features, such as proximity to curves in roads and the height of roadside vegetation. Finally, the suppression of local populations from roadkill itself can also result in a shift of hotspots from one location to another over time [80]. These factors along with our own results suggest that roadkill data alone may not be the best data for validating areas identified as important for connectivity using circuit theory. GPS telemetry data, in contrast, provides hundreds to thousands of data points, from which one can isolate movement data across a variety of landcover types over larger spatial scales, as opposed to strictly proximate to roads [8, 31].

## Comparison to previous studies

Previous studies that analyzed connectivity throughout all of Canada used either large, protected areas as nodes [20, 26] or climate analogs [25, 35]. The former approach is important for helping to address the goal set by parties to the Convention on Biological Diversity to establish well connected systems of protected areas. However, given that only 12.5% of Canada's terrestrial areas are currently protected [28], their approaches could overlook areas important for connectivity in regions with few existing protected areas, including much of southern Canada where most of the country's species at risk reside [72]. Indeed, this is particularly apparent in southern Ontario where our analysis identifies many areas with high probabilities of movement, while the other studies do not. Interestingly, despite the differences in methods, there are many similarities among our results and the findings of the three previous studies. In particular, all three studies identified the Rocky Mountain range in the west as being important for connectivity, as well as around the southern tip of James Bay. In contrast, only our study identified the Chignecto Isthmus as an important area, which is the only terrestrial connection between Nova Scotia and the rest of Canada. Our results are also very similar to those from a study focused on the province of Alberta [6], which used a higher resolution cost surface (100 m) for their smaller study area.

Areas of high current density on our national map also correspond to regions where connectivity conservation initiatives have been established, including Yellowstone to Yukon Conservation Initiative, Algonquin to Adirondacks Collaborative, and the Chignecto Isthmus (Fig 3). This suggests that our map can be used to help support functional connectivity nationally by identifying national priorities and supporting programs such as Canada's national Ecological Corridor Program. We believe the current density map can also be used to identify areas important for functional connectivity within regional initiatives, but we recognize the spatial resolution is likely not fine enough to make decisions on specific land parcels.

Circuit theory maps tend to highlight 'pinch points', or bottlenecks to movement where connectivity is tenuous with few alternative options [30]. This is particularly helpful for identifying areas that should be conserved or restored to maintain connectivity, especially in human-dominated landscapes. Notably, many high current density areas occur in close proximity to areas of high cost, where natural habitat availability is limited [81]. High current density can also occur in high-cost areas that bridge between neighbouring low-cost regions. For these reasons, current density should reflect movement patterns better than the underlying

cost maps [82]. For example, Koen et al. [22] found that current density estimates from an omnidirectional circuit theory model were a better fit to movement data for a variety of taxa than a simple cost surface (input) map, demonstrating that a connectivity map is more than just a habitat map.

Areas with moderate current density do not necessarily illustrate poor connectivity. Indeed, some areas with large swaths of natural habitat (e.g., the Yukon Territory) have many potential pathways where current density is dispersed, and these can make important contributions to overall connectivity even while having lower probabilities of movement per unit area. Regions fortunate to have such areas as options should consider conserving them sooner rather than later, to avoid the creation of pinch points in the future.

## Methodological advances

Our analysis combined and built on several methodological advances in the application of circuit theory to measure connectivity. We applied sensitivity analyses to much larger study areas than had been done before and confirmed that patterns of current density were relatively insensitive to the absolute cost values assigned to land cover types so long as their rank order was maintained [43]. Similarly, our results support previous studies that found that the range of cost values used had an effect on current densities, with broader ranges amplifying 'pinch points', which could help with prioritizing areas important for connectivity [43, 48] (S3C & S3F Fig). We also found that current densities become relatively insensitive to tile sizes above > 1.7 million pixels. Those findings, along with advances in computing power and the Julia implementation of Circuitscape, reinforce the ability to assess connectivity across large study areas. One issue we encountered was that even though we used a buffer width equivalent to 20% of the length of the sides of the tiles, as recommended by Koen et al. (2014), it was not sufficient to eliminate the effects of node placement on current density patterns in two regions. It appears that issue resulted from tile borders being too close to large geographic features (e.g., Great Slave Lake in the Northwest Territories and James Bay in Ontario and Quebec). Fortunately, the issue was easily rectified by using additional tiles centred over the regions with the anomalies.

## Potential limitations

We acknowledge that decisions made during the course our analysis could have affected the results. We classified large lakes as high cost in contrast to Marrec et al. [6], who used a medium cost for their analysis that was restricted to the province of Alberta. We chose the high cost because much of the rest of the country includes many very large lakes, which we believe affect the movement of wildlife most of the year. We recognize that this may not be the case in winter once the lakes have frozen. Similarly, we assigned a high cost to rivers with high flow rates, some of which can freeze in very cold winters. Consequently, patterns of connectivity in winter may differ from our model, suggesting that connectivity may vary seasonally in Canada. We encourage more research on the implications of this idea for land use planning and conservation in Canada.

Although we used the most current land cover data available, we evaluated a snapshot in time and so our analysis does not take into consideration the inevitable continued expansion of the human footprint nor the effects of climate change. More studies like those done by Carroll et al. [25] and Huang et al. [83] are needed to predict connectivity under potential future development and climate scenarios.

When validating our current density map using the roadkill data, we measured the mean current density within a 300 m radius buffer around the road to assess connectivity at the finest

resolution (i.e., immediately proximate to the road) [84, 85]. The buffer size used in such analyses can influence the strength of correlation and hence validation [52]. Further research is required to determine the most appropriate scale to use when comparing current density maps with roadkill data.

At the time of this study, we were unable to obtain wildlife data for the prairies and the far north, which would have strengthened the validation of our current density map. We also recognize that the upstream approach we used undoubtedly does not capture the connectivity needs for all species, nor all complex movement patterns or population processes [10, 86, 87]. In the future, we hope to collaborate with other researchers to compare our omnidirectional, multispecies results to results from species-specific and park-to-park connectivity analyses. In the meantime, we consider that our current density map, which factors in both natural and human-caused restrictions to movement, can be used to accurately identify existing areas important for functional connectivity that should be conserved to help mitigate the negative effects of continued development and climate change on biodiversity.

## Conclusion

We have demonstrated that an upstream modelling approach can be used to characterize functional connectivity for multiple species across a very large study area. We suggest that countries and other large planning areas can use this strategic approach to identify areas important for connectivity to support time-bound initiatives and targets, including connectivity targets in the CBD Post-2020 Global Biodiversity Framework. Our current density map can be used in Canada to help prioritize areas for support by national programs, such as the national Ecological Corridor Program and the 2 Billion Trees Program. The map can also be used for generalized land-use planning at the regional scale, especially in support of planning acts and policies that include requirements for maintaining or restoring connectivity.

## Supporting information

**S1 Table. Land cover layers and their sources used to construct a movement cost layer for Canada.** Movement costs were assigned to anthropogenic layers using our own knowledge but in consultation with members of the Canadian Connectivity Working Group (https://www. conservation2020canada.ca/connectivity). Costs for natural features were assigned based on published data, indicated in the last column, and in consultation with experts (Jodi Hilty, Clayton Lamb). CHF–Canadian Human Footprint, GHF = Global Human Footprint.
(DOCX)

**S2 Table. Movement cost scenarios used to compare across two landscapes with four cost categories.** These scenarios were used to test the sensitivity of mean current densities to absolute values for movement costs and the range of costs. Correlations were calculated among the same 1000 randomly selected cells within pairs of movement cost scenarios using two study areas: a) east coast provinces and b) southern British Columbia.
(DOCX)

**S3 Table. Fixed effects coefficient estimates and model statistics for each species used to validate current density map with GPS telemetry data in western Canada.**
(DOCX)

**S1 Fig. Map showing all 19 tiles used to analyse Canada, with provincial and territorial boundaries in the background.** Black boxes indicate the 17 initial tiles used and red boxes the two additional tiles that were required to address anomalies at the seams. Contains

information licensed under the Open Government Licence–Canada.
(TIF)

**S2 Fig. General locations of independent wildlife data used to validate current density map.** The data included GPS collar data for caribou in British Columbia, moose, wolf, and elk in Alberta; herpetofauna roadkill in Ontario, and moose roadkill in New Brunswick. Contains information licensed under the Open Government Licence–Canada. Public domain animal silhouettes were downloaded from https://beta.phylopic.org.
(TIF)

**S3 Fig. Sensitivity of mean current densities to absolute values of movement costs and the range of costs.** Correlations were calculated among the same 1000 randomly selected cells within pairs of movement cost scenarios using two study areas: a) east coast provinces and b) southern British Columbia. a) Correlations among pairs of cost value scenarios. Mean correlations were 0.79 and 0.84 for the two study areas, respectively. Scenarios 9 and 10 had highest mean correlation values with other scenarios (solid circles denote scenario means, and bars +/- one standard error. Open circles denote correlations of a given scenario with all other scenarios). b) Correlations arranged to display the effect of the range of cost values on current densities. This figure shows the effect of the absolute difference in the range of cost values between pairs of scenarios (log10-transformed) on the Spearman rank correlations. Pairs that include scenarios 9 or 10 are identified by square and triangle symbols, respectively. c) Effect of the range of cost values on current density estimates. Scenarios with broader ranges of costs (i.e., from low to high cost, $\log_{10}$-transformed) result in higher maximum current densities.
(TIF)

**S4 Fig. Effect of tile size on mean current density.** Analysis was conducted on simulated but identical landscapes, to control for composition and spatial distribution of cost values. The same pattern was found for the minimum, maximum, and standard deviation of current densities.
(TIF)

**S5 Fig. Histogram of maximum displacement distance versus number of individuals for each species.** Counts are binned into quartiles, and colour coded. Maximum displacement is the distance from the first recorded location to the location recorded furthest away from that point.
(TIF)

## Acknowledgments

We are very grateful to the researchers that made their wildlife telemetry data available through Movebank.org, to Ontario Nature for the herpetofauna roadkill data, and the Government of New Brunswick for the moose roadkill data. Thank you to Simon Tapper for assisting with data processing. Thanks also to Clayton Lamb, Jodi Hilty, resource conservation staff in the provinces of British Columbia, Ontario, and New Brunswick, as well as member of the Connectivity Working Group for advice. Special thanks to Richard Schuster, Martin-Hugues St-Laurent, and an anonymous reviewer who helped to improve this manuscript.

## Author Contributions

**Conceptualization:** Richard Pither, Jeff Bowman.

**Data curation:** Richard Pither, Paul O'Brien, Angela Brennan, Kristen Hirsh-Pearson.

**Formal analysis:** Paul O'Brien, Angela Brennan.

**Funding acquisition:** Richard Pither, Jeff Bowman.

**Investigation:** Richard Pither, Paul O'Brien, Angela Brennan, Kristen Hirsh-Pearson, Jeff Bowman.

**Methodology:** Angela Brennan, Jeff Bowman.

**Project administration:** Richard Pither.

**Resources:** Paul O'Brien, Angela Brennan, Jeff Bowman.

**Supervision:** Paul O'Brien, Jeff Bowman.

**Validation:** Paul O'Brien, Angela Brennan.

**Visualization:** Paul O'Brien, Angela Brennan.

**Writing – original draft:** Richard Pither, Paul O'Brien.

**Writing – review & editing:** Richard Pither, Paul O'Brien, Angela Brennan, Kristen Hirsh-Pearson, Jeff Bowman.

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
