## [Decision Letter · Decision Letter 0]

27 Jul 2022

PONE-D-22-17312Predicting Areas Important for Ecological Connectivity Throughout CanadaPLOS ONE

Dear Dr. Pither,

Thank you for submitting your manuscript to PLOS ONE. After careful consideration, we feel that it has merit but does not fully meet PLOS ONE’s publication criteria as it currently stands. Therefore, we invite you to submit a revised version of the manuscript that addresses the points raised during the review process.

Please consider and reflect on all points raised by the reviewers (especially noting that reviewer 2 raised a number of concerns on the approach). Each reviewers has provided a well thought out and comprehensive review that will greatly help to improve the manuscript. It should be noted that all reviewers had concerns regarding technological aspects of the manuscript and the clarity of the written text. 

We look forward to receiving your revised manuscript.

Kind regards,

Julian Aherne

Academic Editor

PLOS ONE

Journal Requirements:

2. We noticed you have some minor occurrence of overlapping text with the following previous publication, which needs to be addressed:

- https://www.researchgate.net/publication/342733129_Validation_of_functional_connectivity_modeling_The_Achilles'_heel_of_landscape_connectivity_mapping

The text that needs to be addressed involves the Abstract.

In your revision ensure you cite all your sources (including your own works), and quote or rephrase any duplicated text outside the methods section. Further consideration is dependent on these concerns being addressed.

3. We note that Figures 4, 5, 6, 7, S1, S3 and S4 in your submission contain [map/satellite] images which may be copyrighted. All PLOS content is published under the Creative Commons Attribution License (CC BY 4.0), which means that the manuscript, images, and Supporting Information files will be freely available online, and any third party is permitted to access, download, copy, distribute, and use these materials in any way, even commercially, with proper attribution. For these reasons, we cannot publish previously copyrighted maps or satellite images created using proprietary data, such as Google software (Google Maps, Street View, and Earth). For more information, see our copyright guidelines: http://journals.plos.org/plosone/s/licenses-and-copyright.

a. You may seek permission from the original copyright holder of Figures 4, 5, 6, 7, S1, S3 and S4 to publish the content specifically under the CC BY 4.0 license.  

Additional Editor Comments:

I have received three sets of very comprehensive reviews. All reviewers agree that the manuscript needs substantial revision, two see merit in the approach while the third was not so optimistic. Nonetheless, I have recommended major revisions and I ask the authors to consider all points raised by the reviewers (especially noting that reviewer 2 raised a number of concerns on the approach). Further, all reviewers had concerns regarding technological aspects of the manuscript and the clarity of the written text. Please provide a point-by-point response to each comment.

Reviewers' comments:

Reviewer's Responses to Questions

**Comments to the Author**

1. Is the manuscript technically sound, and do the data support the conclusions?

Reviewer #1: Yes

Reviewer #2: No

Reviewer #3: Partly

2. Has the statistical analysis been performed appropriately and rigorously? 

Reviewer #1: Yes

Reviewer #2: No

Reviewer #3: Yes

3. Have the authors made all data underlying the findings in their manuscript fully available?

Reviewer #1: No

Reviewer #2: Yes

Reviewer #3: Yes

4. Is the manuscript presented in an intelligible fashion and written in standard English?

Reviewer #1: Yes

Reviewer #2: Yes

Reviewer #3: Yes

5. Review Comments to the Author

Reviewer #1: Thank you very much for the opportunity to review manuscript PONE-D-22-17312, “Predicting Areas Important for Ecological Connectivity Throughout Canada”. I very much enjoyed reading this manuscript and think this is an important contribution to our understanding of landscape connectivity for Canada. I have made some comments on several sections of the manuscript below, but in general I think this is a very well developed study and I very much hope to this this published soon.

Movement cost surface (line 172): “In this study, we used both empirical data and expert opinion to construct a movement cost surface for Canada.” Could you please provide details on how expert opinion influenced the cost surface layer, what species this can/is used for, and how the experts determined the values feeding into this study?

Line 213. You mention consultation with experts. Please identify them and provide more details on what information you gathered from them and how.

Circuit theory analysis: Are there ways you could quantify alignment of estimates from overlapping tiles? I am thinking of something along the lines of what folks at Cornell are doing with their AdaSTEM models (e.g. Fink et al. 2020). Not the modelling approach itself, but they way they eventually combine tile or stixel (as they call them) estimates into seamless layers.

Fink, D., Auer, T., Johnston, A., Ruiz‐Gutierrez, V., Hochachka, W. M., & Kelling, S. (2020). Modeling avian full annual cycle distribution and population trends with citizen science data. Ecological Applications, 30(3), e02056.

Fig 1. I am not clear on what the open circles represent. Could you clarify that in the figure legend? Do the open circles represent the other cost scenarios? I think they do, by am not entirely sure.

Reading the corresponding text in the results section I am also not clear why you chose C9. What makes it stand out compared to C1? Could you please provide more detail on your choice and how you chose?

Fig. 4 There is quite the contract in cost values between Canada and the US apparent in Manitoba and moving West. Could you discuss possible implications of that and what causes this difference as well as why you did not try to align datasets across the boarder more?

Also, a red- green color ramp is hard for color blind people to decipher. Could you change the color ramp to something that’s color blind friendly?

Fig. 5. To me this is the main result and figure of this manuscript. I would recommend providing more detail on what readers see on the map in the figure caption. It would be great if the figure (plus caption) could almost be a stand alone product that interested readers could look at and understand easily without needing to read much of the manuscript text.

Validation results: I think the figure numbering is off in this section. I don’t think Fig. 7 shows the validation results for GPS-collar data. If I interpret Fig. 8 correctly, this information is included there. Is that right?

Discussion:

I personally would not go as far as suggest these maps could be used at ‘local-scales’. Applying a Canada wide dataset to local scale connectivity work might be going a bit too far. Regional or landscape scale I can see, but given the input data used for this analysis, I would recommend avoiding ‘local-scales’ analysis as to not give the impression a small land trust could use this information for their planning spanning a couple quarter-sections in say Alberta.

Line 444 “highlight connectivity ‘pinch points’”. I think this is a very important point. Could you discuss the potential consequences of this effect on the results and their interpretation overall?

Line 450+. In this paragraph you mention “We believe the resolution of our map is

sufficient however, to support land management decisions and planning at local-scales, ..”

As mentioned above, I find this framing a bit misleading. I agree that results at the resolution you are using for your map would work for local-scale planning, but I don’t think that all the input data you have used would allow for that. This is the same problem we face in our work at NCC. Our group provides national scale information for regional planning as much as we can, but we do acknowledge and point out that national scale data most often don’t lend themselves well for local-scale planning due to trade-offs between data granularity and spatial scope. That’s a main reason why one of your co-authors (KH-P) set out to refine the global human footprint to Canada. A person in the same group is currently working on refining these data further for use in the province of British Columbia. I would really like to see this important issue discussed in this manuscript.

Line 519+ “Validation of Results”

I very much like your validation of results section and how you used independently collected wildlife data in this manuscript. One thing I am wondering about though is whether correlation indicates causation though? Have you considered including some thoughts on other factors that could contribute to the correlation or how the ‘pinch points’ you mentioned earlier could be correlated with the results you found for herpetofauna?

Line 557+ “Potential limitations”

Thanks very much for including this section. I think its very important that we all discuss potential limitations in our papers. This could be a good place to discuss both the ‘local-scales’ and ‘pinch points’ components that I mentioned earlier in case you don’t think it appropriate to discuss these earlier. I personally think both should ideally be discusses earlier in the text given their importance, but at the very least they should be discussed in this section.

I can’t access the data for review as the OSF repository is currently set to private. Depending on what can be found in the OSF repository the following paragraph might not apply here.

Finally, I would like to include a request to the authors to make their data and code as available as reasonable. I’m a big believer in the FAIR (Findable, Accessible, Interoperable and Reusable) principle for scientific data management and stewardship (e.g. https://www.nature.com/articles/sdata201618), which is why I include a request like this in all my reviews. As an example: my questions related to the statistical tests used and how they were applied could have been answered by looking at the code used in this study. Others could also build on your work once its published.

Best regards,

Richard Schuster

Director of Spatial Planning and Innovation, Nature Conservancy of Canada

Email: richard.schuster@natureconservancy.ca

Reviewer #2: The manuscript takes on the ambitious task of predicting multi-species connectivity across all of Canada. I see the following major issues, the first with the input, and the second with the model choice:

1. Cost surface from expert opinion: The authors claim that they use expert opinion and empirical data to generate cost surfaces. However, their description only points to expert opinion being used for generating costs surfaces. This is highly problematic.

a. The cost surface is the input parameter for connectivity models. They are important, to link species ecology to models, and to ensure that models have realism and utility for on-ground applications. In the past, expert opinion has been widely used, but this practice is fading with studies pointing out the issues with using expert opinion to parameterise models.

Essentially, the study moves from a multi-species perspective (as per the authors' claim), to a study that does not consider species information.

b. All inference on natural barriers etc. to connectivity needs to be removed, as this is simply the input parameters provided by experts into the model.

2. I do not understand why the authors use an omnidirectional approach. Connectivity deals with linkages from a source to a destination. The two ends are important, and are significant. You cannot have connectivity as a cul-de-sac -- going from somewhere to nowhere, or from nowhere to nowhere. Yet, that is what the authors do in their paper.

a. There is a pattern that comes about because habitat is placed the way it is in a landscape. By removing that pattern, you are removing some of the realism of connectivity model and disconnecting it from your landscape.

b. As a result, the findings of the paper largely relate to identifying good quality habitats for biodiversity. I have a feeling the validation of mammalian herbivores is also reflecting the same patters.

I'd strongly suggest modelling connectivity from identified source/destination populations/habitats in the landscape. I am sure the authors have access to this information.

If, like in the northern reaches, there is no clear distinction and animals use the entirety of the landscape, then connectivity is not an issue in this landscape.

3. There are some other methodological clarifications, which I have described in my line-specific comments.

Line-specific comments:

Lines 44-62: This is a lot of text to state that connectivity is important and can be condensed to focus on the goal of this particular exercise. In that sense, while climate change mitigation and migratory species conservation is also relevant, I get the sense that maintaining connectivity across populations and habitats of species is the primary concern here?

Lines 69-82: I am not sure I understand this distinction. Are you suggesting there are two approaches: one that uses species information, and collates resulting maps; the other that uses a sort of least-common-denominator of species movement, i.e., the assumption that species will be impacted by human presence and ignores all other species traits.

Given that, as you say in lines 67-68, species "differ in their movement traits, resource needs and sensitivity to human disturbance", how is this least-common-denominator approach tenable in the face of its mismatch with this theoretical knowledge?

Lines 88-98: You can condense this; Circuit theory based modelling has been in use for a while now.

Also, take a look at: van Moorter et al 2021 Ecography, DOI: 10.1111/ecog.05351, and

Panzacchi et al. 2016, J Appa Ecol. DOI: 10.1111/1365-2656.12386. There is a new approach to modelling animal movement, called Randomised Shortest Path. You may consider using that approach.

Lines 108-110: Under what conditions would you want to model connectivity in general, as opposed to among specific locations? From your introduction, I understood that you wanted to model connectivity among species habitat patches (specific locations).

Lines 111-112: Why are sources and destinations unknown here? Also, I would disagree that the omnidirectional approach assumes uncertainty in sources and destinations. It does carry an assumption about sources and destinations; I think there is some lack of clarity about what this assumption is (locations at the edge of the chosen landscape are sources and destinations?), and this lack of clarity is a problem. Please clarify.

Line 112: What is generalised land use planning activities? There has been a lot of movement towards land use planning that takes into account ecological information on species and ecosystem requirements. If generalised planning activities are those that ignore these requirements, then the use of these is incongruent with advanced movement models, aren't they? Further, I'd note, the solution to the challenge of combining information on multiple species is not to neglect inclusion of such information.

Line 113: It has become available, but why is it useful?

Lines 122-124: I found this finding interesting and surprising, so I went through the paper. The overall correlations may be high, but the locations of high-movement in the cited paper across different relative cost categories changes. This matches with findings from Rayfield et al. 2010, Land.Ecol. DOI: 10.1007/s10980-009-9436-7, who found that least-cost-paths were sensitive, not to absolute values of costs, but to their relative costs. So, assigns costs are important and need considerable thought.

Line 129: By themes, you mean the number of factors influencing resistance? Why is this?

Line 130: There are two things to measure: the 'theme', and its influence on connectivity.

Line 142-143: 'only amongst mostly large protected areas' -- is your contention that they missed a significant proportion of animal habitats or populations?

Lines 144-145: True, but depending on the species, some of them are confined -- in terms of viable source populations -- to PAs. The utility of this is really dependent on the species in question.

Lines 133-146: Why is it important to get this at the national level? This is a huge nation and likely larger than the dispersal range of most species. Likely encompasses multiple conservation landscapes. I am sure there are species that are restricted in range such that they are not present across the country. So what is the relevance or significance or utility even, for a country-wide cross-species connectivity modelling exercise?

Lines 172-173: If you have empirical data, why use expert opinion? Expert opinion has not stood well, nor is it scientifically justifiable, for generating model input parameters.

Lines 174-175: A clarification: You used only those fauna that used natural land uses more, or you found that the majority of fauna used natural land uses more?

If the former, how did you form criteria for inclusion in your analyses?

Lines 175-176: I don't understand this sentence.

Line 183: I don't understand this. You used a buffer outside Canada, and different layers for that buffer? Won't that pose problems in terms of how compatible and directly comparable the layers inside and outside Canada are?

Lines 194-197: Again, I'd point to the fact that relative rankings do not just mean the order, but the quantified relative costs across land uses.

Lines 201-204: I am not sure this approach works. This seems like a pretty superficial comparison. Locations of movement are not all equally important -- pinch points for instance, are more important in terms of inference and prediction than other locations. A correlation coefficient masks these nuances. I'd suggest looking at this deeper in terms of where the variations in rank occur. And while I see why you take ranks -- I agree that the relative and not absolute values make a difference, consider this case: a landscape with large areas of medium movement, some of low movement and some of high movement. A landscape with clearly distinguished low and high movement locations -- rankings will not separate these two scenarios out. You may want to consider scaling current values to be on a comparable scale, instead of ranking.

My second point: it is not the case that the scenario that correlates the most with others is the best scenario.

Line 207: Assigned based on empirical data, you had mentioned. Could you please expand?

Lines 206-219: It seems to me that the costs were assigned based on expert opinion alone. I do not see any details here on how empirical data were used for assigning costs. If so, the authors should state this very clearly. And if so, this is a problem that needs to be addressed.

Lines 239-241: Please expand on these terms.

Lines 243-245: Why is this useful? Taking a step back, how does it fit with dispersal theory? If we go by the assumption that connectivity links between two nodes -- i.e., the nodes are important, and you can't have a stairway to nowhere -- then, where does this mode of connectivity sit? Where is it useful and how does it compare (or not) with connectivity of species or ecosystems between specific nodes.

Lines 257-261: If mammalian herbivores and herpetofauna were the target species, why not formulate cost surfaces for these species/species groups?

Lines 263-273: Were these GPS collar data on dispersing animals or within their home ranges? Current density maps tend to light up 'habitats', due to their low cost. So, if these are of animals within populations / within their habitats, then it will generate high current values, but not because they are useful for movement - it is because they are habitat patches.

Lines 295-297: These are effectively the same cost scenarios, unless you have some cells which are of lower (1) cost. This is because the relative costs (the ratio of costs of any one land use to another) remains the same across the two scenarios. Any difference between the two is likely due to randomness.

Lines 311-312: This points to the fact that there are more 'pinch points' in these cost scenarios.

Lines 293-337: These results are difficult for me to follow or understand, or infer anything from as the cost scenarios are not meaningfully formulated; i.e., there is no ecological hypothesis, or linking to any of the cost scenarios. The paper would be hugely strengthened with this linking of animal ecology and the modelling.

Lines 354-356: But these were your input parameters, so it is not surprising that current densities will be influenced by these parameters?

Lines 367-368: Link this to my previous observation that current densities tend to be high in 'habitats'. This is just picking up suitable areas for animals to 'reside'; they do not necessarily also indicate good connectivity areas.

Lines 367-375: In my reading, this is why the omnidirectional connectivity that ignores known locations of importance for biodiversity (populations), is problematic. It highlights these known locations as important for movement, while they are in reality important habitat. To an extent, is also masks our inference on actual connectivity areas, because they may not show up as clearly as these habitats.

Lines 398--: How many animals were collared, and I hope you included animal ID as a random effect, or tested if this had an impact?

Lines 426-428: Yes, but it was not parameterised by wildlife data.

Lines 433--: These are your assumptions going in, which came from expert opinions, and not data. I would remove all inference about this, unless you can back it with data.

Lines 445-448: This is a good point.

Lines 490-492: Didn't you find that the high cost values were highlighted as the range of cost values increased (as mentioned in your next sentence)? That doesn't support this inference.

Reviewer #3: Dear authors,

I really appreciate reading this manuscript and consider that the topic covered is important, both from a methodological and ecological/conservation perspectives. Building multi-species connectivity models and maps is of prime importance, especially considering the ongoing encroachment of anthropogenic disturbances in the natural/pristine habitats found in Canada. To me, the subject of this manuscript fits the scope of PLoS One, and I consider having the expertise needed to evaluate it, at least some important aspects of it, being implicated in landscape ecology and connectivity research since several years.

Despite my appreciation of the manuscript, and based 4 readings and my expertise/experience, I consider that this manuscript has some weaknesses that can be – I hope/think – corrected in a revised version. Essentially, this is a matter of explaining the novelty a little bit more clearly, revisit the validation process, restructure some passages to focus on the essential/most important results/findings, and readjust the scope to clarify it this is a methodological or an ecological paper (see details below). Writing can be improved (not the language, but the structure of the manuscript), so I consider that more efforts are needed before being able to recommend its acceptance for publication. I’ve listed below the main points/comments that could help clarify the manuscript and hope that these comments will help the author to improve their paper and the Editorial board to evaluate the work done.

1) General comment #1 – Abstract – intro - discussion: Although the authors are clearly recognizing that connectivity is a species-specific concept (line 67), it is unclear if they were modeling structural or functional connectivity. The only mention of functional connectivity is made at line 81 when referring to Koen’s et al. work. This is important, as many of the recent and novel connectivity research is made on functional connectivity, based on animal behaviour using expert knowledge or empirical data. I can understand that it was impossible to model functional connectivity at a community level over such a large study area, but recognizing the differences between structural and functional connectivity is important, at least in the introduction and discussion. In the discussion, clearly positioning the results along this structural – functional connectivity gradient would be interesting.

2) Abstract (line 37): Could be interesting to briefly explain why the validation failed with herpetofauna in the abstract. We have to wait up to the end of the discussion to find out.

3) Introduction (general comment #2): Despite my general comment #1 above, I think that the introduction is quite long (5 ¼ pages). This is not that bad, but I consider that you’ve missed the opportunity to clearly focus on the “lack of knowledge” in the introduction, diluting this with many interesting, but not essential, information that can be moved to the methods (e.g. lines 119-131) or to the discussion (e.g. lines 55-62). This brings me to my second “general comment”: it is unclear in the introduction, as well as elsewhere in the manuscript, if it is a manuscript focusing on methodological advances/development or on ecological/conservation issues. This reflects into the structure of the introduction, with passages referring to the importance of connectivity for the ecology of wildlife in disturbed landscapes (lines 44-53), some statements referring to policy and land-use management (lines 55-62), followed by methodological background (lines 64-131), by the Canadian connectivity status (and importance) (lines 133-146), and finally by the objectives and brief overview of the work done (lines 148-160). Similar bi-modal structure appears in the Results section, with methodological findings (lines 293329, Figs 1-2-3) followed by ecological/landscape ecology results (lines 331-383, Figs 4-7), and back on methodological results (lines 396-414, Figs 8-9). While I can understand that the validation process is important for both methodological and ecological focuses, the first 2/3 of the results are equally treated, which is not essential and can confound the reader (I was questioning myself at several places while reading to figure out what was the real scope, and contribution, of this paper). Obviously, the same bi-modal structure was there in the Discussion too. So my suggestion to the authors is the following: put front – from the introduction to the discussion – the primary focus of your paper. Is it a methodological paper, with clear demonstration that multispecies connectivity models can be built over such a large area, with this spatial resolution (tiles and grain), and with the different cost-scenarios and sensitivity analyses? If so, please diminish the length of the sections focusing on the importance for Canada land-use planning and conservation, from the intro to the discussion; keep some interesting sentences, but shorten these passages. However, I am not convinced this would be the best option, as you just applied and repeated work done by others (among which Jeff Bowman previous papers). So the best option would probably be to focus on this first pan-Canadian, multi-species connectivity modeling exercise, with all the pros and cons noted for the different species during the validation process. If this is the way you want to take, then you could shorten the intro by moving methodological stuff to the methods, shorten the results by sending several paragraphs and figures (1-2-3) to Supplementary material or Appendices, and shorten the discussion by focusing on ecological findings, while reassuring the readers with a short section on the application of the sensitivity analyses. Doing so, you’ll focus on (what I think is) the most important contribution of your paper, in a more balanced manuscript. I let this decision to you (and to the Editors), but this would result in a more interesting – and easier to read and understand – paper.

4) Introduction (lines 148-151) : this is unclear to me what is a spatial grain “sufficient” to support local land management decisions. Could you give some more details?

5) Methods (lines 172-173): This is great that you used both expert opinion and empirical data to construct a movement cost surface for the entire country. But please provide more details, as I wasn’t able to find where “expert opinion” came from, what it is, vs. empirical data. Also, please explain with more details why this is important to combine both source of information, and get back to that in the discussion; is it really a strength? See my main comment regarding the discussion below for more guidelines.

6) Methods (lines 173-177): so you are building a kind of hybrid between a structural and functional connectivity analysis, as you are oversimplifying the heterogeneity of the landcovers (wildlife habitats) for the studied species. This is very “coarse filter”, which is not a real problem considering the number of species modeled but also the impressive spatial scale (as well as the difficulty to gather and put together different sources of information). But this deserves more explanations in the discussion, especially contrasting this with other (one-species only, provincial, territorial, local) connectivity analyses. Again, see my main comment regarding the discussion below.

7) Methods (general comment #3): some decisions made in this section are not supported by (enough) peer-reviewed literature; you can strengthen the confidence of the readers by adding more support to the choice you have made while modeling it. Also, please pay attention to the explanation of the risks associated with the edge-effects (i.e. vignetage; see lines 108, 154, 183-188, 233, 247) while modeling on each tiles. You did great with your analyses, but I am not convinced that a non-initiated reader will fully understand “why” you did what you did.

8) Methods (lines 260-261): Using road kills is not necessarily the best option to validate connectivity, as they occur on roads and roads are often considered as a (fine-scale) high-resistance feature. Could it be possible that the lack of support for the validation of your corridors using herpetofauna roadkill data be linked to the nature of your validation metrics (roadkills) more that to the space-use ecology (home-range size) patterns of these smaller species? This could deserve more explanations in the discussion (just a suggestion).

9) Results (lines 293-322, and Figs 1-2-3): Is this really the most important results of your paper? I know these are needed to understand the following, but these are only methodological stuff, i.e. intermediate steps to reach your goal that was to model connectivity for several species throughout Canada (get back to your manuscript title… nothing about the methodological contributions, just a focus on a pan-Canadian connectivity modeling exercise…). Please considering sending all of it to an Appendix or Supplementary Material.

10) Results (lines 331-337 and Fig. 4): This is obvious, less resistance in the south, more in the north. As are the results regarding connectivity (less in the south, more in the north). Despite this, these two figures (4-5) are very interesting, and deserve publication. But I think that the message they brought could be presented in a more efficient way, i.e. via conservation issues and management imperatives. Please consider revisit the scope of your manuscript, with less methodological findings, you’ll have more room to describe the findings (locally, regionally, nationally) and potential outcomes of your work.

11) Results (validation – lines 398-406): Here I have some doubts regarding the use of GPS-telemetry relocations to validate the connectivity corridors (current density). Looking at lines 400-402, I see the number of data locations, but not the number of animals (as I suspect that you didn’t used 17,444 independent wolves!). So this validation exercise appears to me overly powerful (from a statistical perspective), as the pseudoreplication associated with the same animal having several relocations at one place doesn’t seem to be properly considered in the validation process. Were the relocations belonging to one given animal considered as replicates of this animal, i.e. (explained more clearly, I hope…) that the animal ID is used as a random factor (strata) in the analysis? Also, why using random points vs. GPS relocations? Based on the 95% kernel UD, several random points will end being overlapping real relocations, which would probably decrease the level of contrast between random and GPS points (see figure 8). Why not using 10 or 20 bins of current density to figure out how many relocations (or individuals) were seen in each of these strata of current densities? I am not sure this would be the best way to use your GPS data, but at least it seems to me that it could deserve some more attention given the issues raised above in my comment.

12) Results (lines 402-404): unclear to me: 0.04 is significant, did I miss something?

13) Discussion (general comment): The discussion also could benefit from deciding if the focus of this manuscript is more methodological or ecological. But my main point here is that an important part of the discussion is repeating what was done (methods, e.g. lines 423-431, 450-454, 476-517) or what was found, less on comparing these results (and their interpretation) with the published peer-reviewed literature. Please understand me correctly: I really enjoy your manuscript, but was disappointed to see that the discussion section was not discussing the results as deeply (in details) as what we could expect from this section. Were the corridors found confirming corridors found locally, regionally or provincially in other studies? Were they corresponding to interesting corridors already highlighted for other species than those you have used? Very few references are cited in this section, and I know (having read other papers published by the different coauthors) that you could build a stronger discussion.

14) Discussion (lines 533-534): So what would be the minimum size of a species for which your tiles can be efficient at modeling connectivity? Or what would be the spatial resolution that you recommend to use for herpetofauna?

15) Discussion (lines 557 +): I really appreciate the potential limitation section.

16) Conclusion (general comment): The conclusion is mostly repeating things that we have already read in other sections of the manuscript. Could you revisit this section to add some recommendations? Potential outcomes of the use of your connectivity map to contribute to the conservation of different species and wildlife habitats?

17) Figures 1-2-3: maybe it’s just the way the figures were joined to the manuscript, but they can be smaller, all presented into one figure with several panels.

18) Figure 9: Sorry but I cannot see the positive correlation here… the relationship appears to me negative. Have you tried to put a GAMM or a non-linear curve on this group of points? I am surprised that this relationship is significant; please explain more in details these opposite perspectives between your text (positive relationship) and your figure (negative relationship, although strongly driven by a few points).

Thank you for this promising manuscript, and good luck with the revision.

6. PLOS authors have the option to publish the peer review history of their article (what does this mean?). If published, this will include your full peer review and any attached files.

Reviewer #1: **Yes: **Richard Schuster

Reviewer #2: No

Reviewer #3: No

---

## [Author Response · Author response to Decision Letter 0]

16 Sep 2022

Our response to the Editor and the reviewers is in an attached document

---

## [Decision Letter · Decision Letter 1]

12 Dec 2022

PONE-D-22-17312R1Predicting Areas Important for Ecological Connectivity Throughout CanadaPLOS ONE

Dear Dr. Pither,

Thank you for submitting your manuscript to PLOS ONE. After careful consideration, we feel that it has merit but does not fully meet PLOS ONE’s publication criteria as it currently stands. Therefore, we invite you to submit a revised version of the manuscript that addresses the points raised during the review process.

Specifically, Reviewer 3 has noted a number of minor technical revisions to help improve the manuscript; further, Reviewer 2 had wider concerns regarding the use of expert opinion and omni-directional approaches that ignore 'nodes'.

We look forward to receiving your revised manuscript.

Kind regards,

Julian Aherne

Academic Editor

PLOS ONE

Journal Requirements:

Additional Editor Comments:

I have received reports from three reviewers, which on balance support the publication of this manuscript. However, Reviewer 3 has noted a number of minor technical revisions to help improve the manuscript; further, Reviewer 2 had wider concerns regarding the use of expert opinion and omni-directional approaches that ignore 'nodes'. I look forward to seeing a revised manuscript that addresses the comments from the reviewers.

Reviewers' comments:

Reviewer's Responses to Questions

**Comments to the Author**

1. If the authors have adequately addressed your comments raised in a previous round of review and you feel that this manuscript is now acceptable for publication, you may indicate that here to bypass the “Comments to the Author” section, enter your conflict of interest statement in the “Confidential to Editor” section, and submit your "Accept" recommendation.

Reviewer #1: All comments have been addressed

Reviewer #2: (No Response)

Reviewer #3: All comments have been addressed

2. Is the manuscript technically sound, and do the data support the conclusions?

Reviewer #1: Yes

Reviewer #2: No

Reviewer #3: Yes

3. Has the statistical analysis been performed appropriately and rigorously? 

Reviewer #1: Yes

Reviewer #2: No

Reviewer #3: Yes

4. Have the authors made all data underlying the findings in their manuscript fully available?

Reviewer #1: Yes

Reviewer #2: Yes

Reviewer #3: Yes

5. Is the manuscript presented in an intelligible fashion and written in standard English?

Reviewer #1: Yes

Reviewer #2: Yes

Reviewer #3: Yes

6. Review Comments to the Author

Reviewer #1: Thank you very much for your attention to detail in addressing the comments from all three reviewers in this revised version of your manuscript. I have paid particular attention to the responses to my original comments and am happy with all responses you have provided.

As for the other reviewer’s comments, I have read over them as well and even though I might not agree with some of the points the reviewers made I do appreciate the attention and care you have used to address each and every comment.

I have no further comments at this point and am happy to recommend publication of this manuscript.

Best regards,

Richard Schuster

Director of Spatial Planning and Innovation, Nature Conservancy of Canada

Email: richard.schuster@natureconservancy.ca

Reviewer #2: I thank the authors for their responses to my comments. I have some follow-up comments. And I apologise for not being more supportive, but I feel that my biggest concerns were not adequately addressed, or perhaps not fully understood. There are three major issues: (1) the use of expert opinion; (2) related to this, the authors insistence that relative costs do not matter, when there is evidence to show that they do; and (3) use of omni-directional approaches that ignore 'nodes'.

I'd like to say that I appreciate the scale of the landscape, and that there can be substantial practical application for such an effort, with government policy backing. However, this makes it all the more important to have good science, and good connectivity maps that are based on ecological knowledge and empirical data.

I have provided my detailed comments below but before that, I can provide some pointers on how to address these:

1. Use of expert opinion: State from the beginning that the authors take a structural approach to connectivity (different from an upstream approach); and that the authors expect such an approach to be useful for a large number of species that are dependent on natural ecosystems / specialists to natural ecosystems. Clearly state (even in the abstract) that the cost was based on expert opinion, while the movement model results were validated with empirical data on such species (i.e., clarify that empirical data was only used for validation).

2. Relative costs vs ranks: Clearly state throughout the MS that relative ranks matter (ie., what has been referred to as range of values). Run the sensitivity test against high-movement / pinch-point areas alone (e.g., correlation of top 10% movement areas -- or areas which would be identified as corridors, or overlap of identified high-priority connectivity areas across scenarios). Include this as a caveat to your manuscript.

3. Omnidirectional connectivity: Have a clear explanation for what this means. As of now, you have provided corridors identified for a hypothetical scenario where all animals are at the boundary of Canada. Please state -- what would this signify for the species ecologically and/or from a conservation perspective. Include a caveat that if data on animal distribution were available, it would be an improvement to add such information.

Detailed comments:

1. Use of expert opinion:

Response 1- I accept that there can be two approaches, one where connectivity is modelled for each species and then combined, and one where the 'resistance' is combined across all species. My point was not this.

My point was that you use expert opinion to parameterise (multi-species) resistance. Expert opinion is problematic for accurately reflecting the resistance of a single species; it is going to be as, or much more, difficult to reflect the needs of multiple species.

You note in lines 142-143 that: "Cost surfaces can be constructed using expert opinion or empirical data". This is true, but, there has been a move away from expert opinion and specific calls for such a move.

With this -- and because your expert opinion was based on how far a location is from 'natural-ness', I would suggest that your approach is closer to a structural view of connectivity than a functional one. There is no issue with using a structural view, but your manuscript needs to accurately and truthfully reflect this. You can clearly say that you have used a structural approach - validated with locations of multiple species - that assumes that this will be relevant for a set of species that are dependent on natural environments.

Note: A structural approach cannot be interchangeably used with an upstream approach, in its absolute sense. An upstream approach speaks to collating data across species at the 'resistance' parameterisation step. You assume that the expert opinion approximates these data. Thus, even in an upstream approach, empirical data has a place, has value, and your work replaces that with expert opinion.

This is fine (not ideal), but you need to be very clear about it.

And you do have to discuss this as a caveat in the discussion.

Also, Response 4 & 5: Thank you for clarifying the upstream and downstream approaches. Throughout, I'd like to clarify that this bit of your approach is now clear to me. What I (was and) am interested in is: within your upstream approach, how you incorporated species-movement information or empirical data.

When combining species connectivity maps, there are multiple approaches: you could take the locations that matter to the most number of species; you could sum current (ie., choose locations that contribute the most across species); you could choose pinch points for a set of species.

What would be your associated approach here. Are you seeing this as taking least-common-denominator of species requirements for movement (natural-ness), or do you see this as encompassing the maximum number of species, or do you see this as encompassing there most sensitive species?

Side note: I'm a bit confused about your statement: "there is no common set of species that occur across all of Canada". Of course there won't be, but there doesn't have to be. A corridor will serve more species, where there are more species; and it will not be useful in a landscape with (as a caricature example) there are no species. It is meaningless to have a jaguar corridor in Montana -- again a caricature example - or even in Northern California.

All of this makes my worry about the practical applications of this output. Assuming that this will go into some sort of corridor demarcation or policy, how will it account for the lack of the realism in terms of not reflecting underlying species range limits and distributions; in terms of not specifying what is being connected where.

From the author response: "This is important in the context of providing advice for applied land-use planning, where details about node placement may vary from application to application." Could you please explain why node placement would vary? Do you mean from species to species?

2. Relative cost rankings:

Response 12: The absolute cost values do not matter. But the relative cost, in addition to the rank order of matrix types matters and this has been shown previously (i.e., ratio vs ordinal data).

Rayfield et al. (2010) found that: "spatial location of least-cost routes was sensitive to differences in relative cost values assigned to landcover types".

From Bowman et al. (2020): "We also found that correlations were more variable when the range of cost values in a map was high". Also Fig. 3.

Which means that statements such as this the one below (lines 168-170) made in the MS are misleading and patently wrong.

(lines 168-170: "Current density maps have low sensitivity to the absolute value of the costs assigned to land cover types so long as the relative ranking of the types is maintained"

Not only do you need to maintain the relative ranks, you also need to retain the ratio of costs across categories. This has been shown by both citations, see above.

The authors themselves go on to say: (lines 172-174) "However, the range of cost values (e.g., 1 to 3 versus 1 to 1000) does appear to have a small effect on the pattern of current densities". This is the effect of the relative values of costs, and it is not a small effect.

And, they agree with this in responses 31 and 32. Yet, it is not reflected in the methods section, and methodological approach of the MS.

The authors say they don't understand what I meant with this statement: "the locations of high-movement in the cited paper across different relative cost categories changes" There are two implications in this statement:

a. Relative costs matter (not just ranks)

b. It is not the average current densities that are of most importance, ecologically speaking, it is the location of high-current densities. The location of high-current densities, and the intensity of current in these locations, changed when Bowman et al. (2020) changed relative ranks (range of ranks, as per the article), and in Rayfield et al. (2010).

But there is one additional implication, for response 23(b). Locations of high-movement are significantly more important, but proportionally of a smaller area in the landscape. If you do a rank correlation, you will not (a) detect if there is significant difference between high and low movement areas -- a big impact of changing relative ranks; and (b) your results will be overwhelmed by med-low movement areas that are proportionally more represented in the landscape (ie., there is a large area covered by low-med movement, and a small area covered by high-movement). Hence, my reflection that this comparison is superficial, and will mask differences across your cost scenarios.

3. Omni directional connectivity:

Response 2: The authors say that modelled connectivity patterns are dependent on node placement. But shouldn't it be? This is reflective of a reality wherein 'true' connectivity patterns will depend on where animals are present. If they are absent from an area, there will be no connectivity to that area.

Note: This may or may not overlap with where Protected areas are; that does not determine the feasibility or applicability of this approach.

From your response: "current density patterns are biased by the placement of nodes (i.e., the same landscape will produce widely differing outputs depending on node placement)". As they should - if they weren't biased to node placement (reflecting our assumptions of where animals are), there would be a problem with our model.

In that, I do not understand what movement, not dependent on node placement, means or ecologically signifies (because I cannot find the analogue for it in 'true' ecological patterns). Therefore, I do not know what the model will produce, and I do not know what to equate the model output with in terms of ecological processes. The authors need to clarify this, but have not.

In response 28, the authors say: "a detailed exploration of the theoretical underpinnings of the method are beyond the scope of our current paper" and I agree, but the authors will need to at least be clear on, and state, what their results mean ecologically.

I'd also point out that omnidirectional connectivity does not remove assumptions on node placement; they just move the nodes. Ie., you are assuming that animals want to move from the buffer of the map (where nodes are placed).

Reviewer #3: Good morning everyone,

I want to warmly thank the reviewers for their open-mindedness relative to the comments we all raised after reading the original version of their manuscript. I (Reviewer #3 on the first round) am satisfied with most of the changes made and the way the manuscript has been revisited. This is clearer now, more straightforward, less balancing between an ecological story and a methodological paper; I really think that the manuscript is stronger this way.

Nevertheless, I still have some minor concerns that may deserve some attention before going forward in the publication process. I have listed them below, hoping that they could be interesting for the authors in the finalization of their revised manuscript, and for the Editorial board to take a decision regarding this promising paper.

1) General comment #1 – structure (mainly discussion): Excellent revision of the structure of your manuscript, congratulations! The flow of arguments is clearer now, thank you. Nevertheless, I suggest adding another subtitle in the discussion, ~line 487, as the paragraph here (lines 487-511, maybe lines 487-529) seem to belong to another “section” compared to lines 432-485). While the lines 487-511 and 524-529 refer to the location of the corridors, lines 513-522 could be moved to the limitations or even more interestingly to the methods. In contrast, lines 433-485 explain more in details the strength of the exercise and the validation process.

2) Methods (lines 260-262): I suggest adding a few references in support to this decision, as the 10% of the GPS data sample for each individual seems quite a random decision and a random threshold.

3) Methods (lines 294-296 and 303-304): Again, please provide some support to the decision of using a 300-m radius buffer for the herpetofauna roadkill data, and a 900 x 900m windows along roads for the moose roadkill data. In a recent paper, we (Laliberté & St-Laurent 2020, Landscape and Urban Planning) have shown that the radius buffer could influence (slightly, but this is a real effect) the strength of the correlation during the validation process. I suspect that maybe testing different buffer sizes would have influenced, even very slightly, the validation strength.

4)Results (lines 316-318): I am not convinced that this reason (“… because that value has been used in previous studies”) is sufficient (or appropriate). CircuitScape cannot deal with null values, that’s why we usually add “+1” to all resistance values.

5) Results (line399-401 and Fig. 7): I think this could be important to bring to the attention of the authors that even if a positive, significant correlation has been found between current density and moose roadkill, this relationship is quite weak (see the very low Pearson r on Fig. 7). Looking at the figure suggest that this relationship is very “noisy” with a lot of variation in the number of roadkills for the same current density. This would open the opportunity to discuss the reliability of the different validation metrics in your discussion (as we did in Laliberté & St-Laurent 2020), and to identify potential other factors that may mask this expected relationship.

6) Discussion (lines 462-463): Along with my comment just above, here you have to spend a little bit more time explaining why the different validation metrics performed differently, especially the moose roadkills. I suggest to view your result critically even if the relationship follows what was expected.

7) Discussion (lines 483-485): please add some examples, supported by the literature, for these “other factors”.

8) Discussion (Limitations): I REALLY enjoyed your section about potential limitations, nice job. You might have to consider talk about the “vignetage” issue that may be at play in some maps (see for examples Fig. 4a-b-c), but this issue needs more explanation in the methods (that’s why you overlapped some tiles…), especially regarding official boundaries (e.g. USA vs. Canada).

I sincerely hope that the comments listed above will be helpful for you, congratulation again for this interesting manuscript, I will use and cite it in my own research in the future.

Best regards,

Martin-Hugues St-Laurent

7. PLOS authors have the option to publish the peer review history of their article (what does this mean?). If published, this will include your full peer review and any attached files.

Reviewer #1: **Yes: **Richard Schuster

Reviewer #2: No

Reviewer #3: **Yes: **Martin-Hugues St-Laurent

---

## [Author Response · Author response to Decision Letter 1]

12 Jan 2023

RESPONSES

TO ACADEMIC EDITOR

Thank you for the opportunity to further improve our manuscript. We have made more revisions in response to the reviewers very helpful comments, as outlined further below. We truly appreciate the time the reviewers have contributed to this process, resulting in what we believe is a much stronger manuscript. 

We have been able to respond to all the review comments, which were mostly straightforward. We disagree with R2’s opinion regarding omnidirectional methods, which we tried to address in the first round of responses. This method is frequently used in the connectivity conservation literature (outlined below) as it helps to address the biases that can occur using source-to-destination connectivity methods, which we describe in lines 122 – 130 and 223 – 228 (all line numbers correspond to those in “Revised_Manuscript_with_Track_Changes.doc” when viewed using ‘All Markup’ in Track Changes). We also note that it is unrealistic to expect that all source and destination nodes would be known, especially for multiple species throughout Canada, and for the many different applications of connectivity analyses. 

As mentioned before, the omnidirectional connectivity method was first published in PLOS One (Pelletier et al. 2014) as well as in Methods in Ecology and Evolution (Koen et al. 2014) and has since been used in numerous other studies (a combined 287 citations according to Google Scholar; with specific examples included in our response to the first review). Nevertheless, we indicate towards the end of the Discussion (lines 758 - 760) that it would be desirable to compare the results of our omnidirectional analysis to results from species-specific and park-to-park connectivity analyses. We have started to reach out to other researchers for more species’ data and model results and hope that their use in a future analysis will help address Reviewer 2’s skepticism regarding the value of the omnidirectional method. 

With respect to R2’s concern about the use of expert opinion, we acknowledge that in general, it is less desirable than empirical data. It is important to appreciate that collecting movement data for many individuals of many species typically incurs massive financial costs and logistical challenges, which is why one of the stated objectives of our research was to determine whether a simpler approach could be used to model multi-species connectivity across Canada. Therefore, using expert opinion was an explicit goal of our study. As we describe in the manuscript, we used only four movement cost values (more would lead to greater uncertainty in the results) and the values we assigned were consistent with other studies (references: 6, 15, 17–20, 22, 26, 31–34) and based on published studies of species’ biology, which we cite (6, 34, 39, 40-42, 45-49). Most importantly, our resulting current density map was validated with independent wildlife data. 

Although we understand R2’s perspective on structural versus functional connectivity, we also agree with the reviewer that these approaches exist on a gradient (see Reviewer 2’s comment that “your approach is closer to a structural view of connectivity than a functional one”). As R3 also noted in the Round 1 comments, connectivity models are usually on a “gradient” between the simplified dichotomous concepts of structural and functional connectivity. The very act of parameterizing a movement cost layer based on generalized species movement patterns adds some degree of functionality. Ultimately, however, the fact that independent wildlife movement data was found significantly more often in areas with high current densities compared to available but unused areas demonstrates that our model was indeed predicting areas of functional connectivity. We were intrigued by R3’s point (round 1) that it would be very interesting to further discuss the structural - functional gradient, but we feel that it merits its own manuscript.

We have provided responses to each reviewer comment below, and we look forward to your further review. 

Two references related to the CBD Post-2020 Biodiversity Framework were replaced with a single reference to support changes in the text, which now refers to the adoption of the Framework in December.

Thank you very much,

Richard Pither

Reviewer #1: Thank you very much for your attention to detail in addressing the comments from all three reviewers in this revised version of your manuscript. I have paid particular attention to the responses to my original comments and am happy with all responses you have provided.

As for the other reviewer’s comments, I have read over them as well and even though I might not agree with some of the points the reviewers made I do appreciate the attention and care you have used to address each and every comment.

I have no further comments at this point and am happy to recommend publication of this manuscript.

Best regards,

Richard Schuster

Director of Spatial Planning and Innovation, Nature Conservancy of Canada

Email: richard.schuster@natureconservancy.ca

Reviewer #2: I thank the authors for their responses to my comments. I have some follow-up comments. And I apologise for not being more supportive, but I feel that my biggest concerns were not adequately addressed, or perhaps not fully understood. There are three major issues: (1) the use of expert opinion; (2) related to this, the authors insistence that relative costs do not matter, when there is evidence to show that they do; and (3) use of omni-directional approaches that ignore 'nodes'.

I'd like to say that I appreciate the scale of the landscape, and that there can be substantial practical application for such an effort, with government policy backing. However, this makes it all the more important to have good science, and good connectivity maps that are based on ecological knowledge and empirical data.

I have provided my detailed comments below but before that, I can provide some pointers on how to address these:

1. Use of expert opinion: State from the beginning that the authors take a structural approach to connectivity (different from an upstream approach); and that the authors expect such an approach to be useful for a large number of species that are dependent on natural ecosystems / specialists to natural ecosystems. Clearly state (even in the abstract) that the cost was based on expert opinion, while the movement model results were validated with empirical data on such species (i.e., clarify that empirical data was only used for validation).

RESPONSE 1. We sincerely appreciate the time the reviewer has taken to review our manuscript. We have addressed several comments in both rounds of reviews and are confident the manuscript has improved as a result. For responses to the pointers, please see details below. For this specific pointer, please see detailed Response 4.

2. Relative costs vs ranks: Clearly state throughout the MS that relative ranks matter (ie., what has been referred to as range of values). Run the sensitivity test against high-movement / pinch-point areas alone (e.g., correlation of top 10% movement areas -- or areas which would be identified as corridors, or overlap of identified high-priority connectivity areas across scenarios). Include this as a caveat to your manuscript.

RESPONSE 2. Please see detailed Responses 10 - 13.

3. Omnidirectional connectivity: Have a clear explanation for what this means. As of now, you have provided corridors identified for a hypothetical scenario where all animals are at the boundary of Canada. Please state -- what would this signify for the species ecologically and/or from a conservation perspective. Include a caveat that if data on animal distribution were available, it would be an improvement to add such information.

RESPONSE 3. Please see detailed Responses 14 - 17.

Detailed comments:

1. Use of expert opinion:

Response 1- I accept that there can be two approaches, one where connectivity is modelled for each species and then combined, and one where the 'resistance' is combined across all species. My point was not this. My point was that you use expert opinion to parameterise (multi-species) resistance. Expert opinion is problematic for accurately reflecting the resistance of a single species; it is going to be as, or much more, difficult to reflect the needs of multiple species.

You note in lines 142-143 that: "Cost surfaces can be constructed using expert opinion or empirical data". This is true, but, there has been a move away from expert opinion and specific calls for such a move. 

RESPONSE 4: Although we agree that empirical data has obvious advantages over expert opinion, one of the objectives of our study, as stated in the Introduction, was to determine whether a simpler approach could be used to model multi-species connectivity across Canada. We have updated the abstract (line 29) and the last paragraph in the Introduction to explicitly state our use of expert opinion (line 138).

Note: all line numbers correspond to those in “Revised_Manuscript_with_Track_Changes.doc” when viewed using ‘All Markup’ in Track Changes.

—---------------------------

With this -- and because your expert opinion was based on how far a location is from 'natural-ness', I would suggest that your approach is closer to a structural view of connectivity than a functional one. There is no issue with using a structural view, but your manuscript needs to accurately and truthfully reflect this. You can clearly say that you have used a structural approach - validated with locations of multiple species - that assumes that this will be relevant for a set of species that are dependent on natural environments.

RESPONSE 5: As implied by the reviewer, connectivity models can fall on a gradient between the structural and functional connectivity. The steps of parameterizing a movement cost layer based on generalized species movement patterns and using a random walk algorithm to model individual movement adds some degree of functionality. More importantly, our resulting current density map was validated with independent data for several wildlife species. Specifically, GPS telemetry data from individuals that moved longer distances (wolves, caribou, moose, elk) were found significantly more often in locations of high current densities than unused but available locations. Therefore, we consider that our map predicts functional connectivity for select species. We recognize however, that the gradient exists between these model types and have tried to make that clear in the discussion. We removed the word functional from the first sentence of the discussion and have added a new paragraph where we briefly address these concepts (Lines 580 - 586):

● “Use of an upstream, generic species approach[9] and a human footprint could be considered a model of structural connectivity. However, we attempted to include elements of functional connectivity by ranking our movement costs based on published studies of species’ biology and employing a random walk algorithm to model movement. Furthermore, the predictions of our model were supported by independent movement data for some species. Indeed, we consider that there is a gradient between the concepts of structural and functional connectivity, and that our study has aspects of both types”

—----------------------------------------------------

Note: A structural approach cannot be interchangeably used with an upstream approach, in its absolute sense. An upstream approach speaks to collating data across species at the 'resistance' parameterisation step. You assume that the expert opinion approximates these data. Thus, even in an upstream approach, empirical data has a place, has value, and your work replaces that with expert opinion. This is fine (not ideal), but you need to be very clear about it. And you do have to discuss this as a caveat in the discussion.

RESPONSE 6: We agree that those terms are not interchangeable and have not used them as such. As we noted in the manuscript, we were following the Wood et al. (2022) definition, which states : “Upstream methods, such as species agnostic or generic species approaches, require substantially less species-specific data and computational resources” (our emphasis with bold text). We described this generic species approach in lines 85 - 99. We have now revised our stated objective in the Introduction to clarify that we used expert opinion (line 138). Our abstract already stated that we “... developed a movement cost layer with cost values assigned to anthropogenic land cover features and natural features based on their known and assumed effects on the movement of terrestrial, non-volant fauna.” Importantly, our resulting current density map was validated with independent data for several wildlife species. 

—------------------------------------------------------

Also, Response 4 & 5: Thank you for clarifying the upstream and downstream approaches. Throughout, I'd like to clarify that this bit of your approach is now clear to me. What I (was and) am interested in is: within your upstream approach, how you incorporated species-movement information or empirical data. When combining species connectivity maps, there are multiple approaches: you could take the locations that matter to the most number of species; you could sum current (ie., choose locations that contribute the most across species); you could choose pinch points for a set of species. What would be your associated approach here. Are you seeing this as taking least-common-denominator of species requirements for movement (natural-ness), or do you see this as encompassing the maximum number of species, or do you see this as encompassing there most sensitive species?

RESPONSE 7: Yes, we essentially used the common-denominator of species’ movement behaviour in response to anthropogenic and natural land cover features. We stated that “we modelled connectivity for terrestrial, non-volant fauna that use and move across natural land cover more successfully than anthropogenic land cover types by assigning higher costs to landscape elements with higher degrees of anthropogenic disturbance.” (Line 158 - 160). Meaning we tried to encompass the maximum number of species that are sensitive to anthropogenic disturbance. Although we did not use animal movement data to model and parameterize the movement cost layer, we constructed the layer and assigned cost values consistent with other studies (references: 6, 15, 17–20, 22, 26, 31–34) and based on published studies of species’ biology, which we cite (6, 34, 39, 40-42, 45-49) in a few locations, including lines 148-149. And we did so to see if such an approach would predict areas important for connectivity that would correspond with observed animal movement patterns. This is explained in lines 75 - 85 and 539 - 541. Furthermore, we explicitly state that our analysis was designed to predict connectivity for species that “move across natural land cover more successfully than anthropogenic land cover types” (lines 207 - 211). Importantly, our approach was validated using independent wildlife data, which indicates support for our approach.

—-------------------------------------------------------

Side note: I'm a bit confused about your statement: "there is no common set of species that occur across all of Canada". Of course there won't be, but there doesn't have to be. A corridor will serve more species, where there are more species; and it will not be useful in a landscape with (as a caricature example) there are no species. It is meaningless to have a jaguar corridor in Montana -- again a caricature example - or even in Northern California. All of this makes my worry about the practical applications of this output. Assuming that this will go into some sort of corridor demarcation or policy, how will it account for the lack of the realism in terms of not reflecting underlying species range limits and distributions; in terms of not specifying what is being connected where.

RESPONSE 8: The manuscript states that the goal of this project was to test the upstream approach using a generalized movement cost layer, with costs assigned to reflect those for species whose movement patterns are known to be negatively affected by anthropogenic features. We also state that the “upstream approach we used undoubtedly does not capture the connectivity needs for all species, nor all complex movement patterns or population processes …” (Lines 756 - 758). We agree with Reviewer 2’s implication that our current density layer alone should not be used to demarcate corridors. Indeed, our current density layer is being used already in combination with other data layers and local knowledge by governments and agencies to prioritize conservation decisions and avoid the identification of meaningless corridors. But we consider it useful to have a seamless map produced in a consistent way across an area as large and diverse as Canada, where a common currency (current density) can be considered among other values.

—----------------------------------------------------------------------------

From the author response: "This is important in the context of providing advice for applied land-use planning, where details about node placement may vary from application to application." Could you please explain why node placement would vary? Do you mean from species to species?

RESPONSE 9: We meant that because land-use planning involves multiple objectives and species, node placement for different objectives would most likely vary. E.g., Different nodes would be required when identifying areas important for connectivity between two specific protected areas, identifying locations for wildlife passages to reduce road mortality in areas with no protected areas, identifying potential corridors between disjunct populations for an endangered species, or for recolonization of habitats following local extinctions. We also reiterate that in many situations, sources and destinations are not known, especially for multiple species throughout much of Canada (lines 131 - 133).

—--------------------------------------------------------------------

2. Relative cost rankings:

Response 12: The absolute cost values do not matter. But the relative cost, in addition to the rank order of matrix types matters and this has been shown previously (i.e., ratio vs ordinal data).

Rayfield et al. (2010) found that: "spatial location of least-cost routes was sensitive to differences in relative cost values assigned to landcover types". From Bowman et al. (2020): "We also found that correlations were more variable when the range of cost values in a map was high". Also Fig. 3.

Which means that statements such as this the one below (lines 168-170) made in the MS are misleading and patently wrong. (lines 168-170: "Current density maps have low sensitivity to the absolute value of the costs assigned to land cover types so long as the relative ranking of the types is maintained" Not only do you need to maintain the relative ranks, you also need to retain the ratio of costs across categories. This has been shown by both citations, see above.

RESPONSE 10. We think there is some confusion about terms here. It appears that the reviewer uses the phrase “relative costs” to mean much the same thing as the ratio of the costs, whereas in the statement at lines 183 - 185 that “Current density maps have low sensitivity to the absolute value of the costs assigned to land cover types so long as the relative ranking of the types is maintained." We use relative to indicate rank order of costs relative to one another. To clarify this, we have now removed the word relative and rephrased this sentence as “Current density maps have low sensitivity to the absolute value of the costs assigned to land cover types so long as the rank of the types is maintained". This was the conclusion of the paper by Bowman et al. (2020) after a simulation study (the lead author of that paper is an author of the current research being reviewed here). There is a small effect of the range of costs and of the ratio of the costs, but the rank order of the costs is overwhelmingly the most important effect. Therefore, sensitivity to these other effects is relatively lower (i.e. low sensitivity). This is the simple, important point we are trying to make. The mean correlation in current density values across all scenarios with different relative cost values but maintaining rank order in the Bowman et al. (2020) paper varied from 0.87 to 0.90 depending on fragmentation level. Therefore, Bowman et al. (2020) concluded sensitivity to absolute costs was low. We also note that the Rayfield et al. (2010) paper is a bit less relevant to the questions here because it deals with sensitivity of least cost paths which have different properties due to the route optimization.

—--------------------------------

The authors themselves go on to say: (lines 172-174) "However, the range of cost values (e.g., 1 to 3 versus 1 to 1000) does appear to have a small effect on the pattern of current densities". This is the effect of the relative values of costs, and it is not a small effect.

RESPONSE 11. Please see Response 10. The mean correlation in current density values across all scenarios with different relative cost values but maintaining rank order in the Bowman et al. (2020) paper varied from 0.87 to 0.90 depending on fragmentation level. We consider the remaining unexplained variation to be a small effect. Perhaps this is semantics, but it is at least a much smaller effect than the variation due to the rank order of the costs.

—--------------------------------------

And, they agree with this in responses 31 and 32. Yet, it is not reflected in the methods section, and methodological approach of the MS.

RESPONSE 12. We explicitly address this variation due to different cost scenarios using a sensitivity analysis. See S2 Table and S2 Figure for details. This is the very purpose of the sensitivity analysis, which found that mean correlations across 10 different cost scenarios with the same rank order varied between 0.79 and 0.84 depending on location. Unexplained variation was due to relative differences in the costs. We selected a highly correlated scenario as the basis for our model. 

—--------------------------------------

The authors say they don't understand what I meant with this statement: "the locations of high-movement in the cited paper across different relative cost categories changes" There are two implications in this statement:

a. Relative costs matter (not just ranks)

b. It is not the average current densities that are of most importance, ecologically speaking, it is the location of high-current densities. The location of high-current densities, and the intensity of current in these locations, changed when Bowman et al. (2020) changed relative ranks (range of ranks, as per the article), and in Rayfield et al. (2010).

But there is one additional implication, for response 23(b). Locations of high-movement are significantly more important, but proportionally of a smaller area in the landscape. If you do a rank correlation, you will not (a) detect if there is significant difference between high and low movement areas -- a big impact of changing relative ranks; and (b) your results will be overwhelmed by med-low movement areas that are proportionally more represented in the landscape (ie., there is a large area covered by low-med movement, and a small area covered by high-movement). Hence, my reflection that this comparison is superficial, and will mask differences across your cost scenarios.

RESPONSE 13: The reviewer made this same point in the previous review and suggested that we scale current density values prior to correlating them to address the perceived problem. We did this in response to the reviewers comment last time.

From our response during the last review: Response 23. We prefer to use Spearman correlations in the manuscript because they are more robust than Pearson correlations to potential violations of spatial non-independence among the samples. However, we have also followed the reviewer’s suggestion here, and compared scenarios using mean-centred data (z scores) with Pearson correlations, which also demonstrated high correlations across all scenarios, supporting the conclusions we reached with the Spearman correlations (Maritimes: Spearman, mean rs = 0.79, range = 0.39-1.0; Pearson of standardized values, mean r = 0.70, range = 0.25-1.0; BC: Spearman, rs = 0.87, 0.63-1.0; Pearson, mean r = 0.79, 0.58-1.0). Therefore, after following the reviewer’s suggestion, we remain confident that current density tends to be highly correlated across cost scenarios provided that the rank order of the costs is maintained.

We remain satisfied that our comparison is not superficial, and that in fact, current density values tend to be highly correlated across scenarios provided that the rank order of the costs has not changed. 

—-------------------------------------

3. Omni directional connectivity:

Response 2: The authors say that modelled connectivity patterns are dependent on node placement. But shouldn't it be? This is reflective of a reality wherein 'true' connectivity patterns will depend on where animals are present. If they are absent from an area, there will be no connectivity to that area. Note: This may or may not overlap with where Protected areas are; that does not determine the feasibility or applicability of this approach.

RESPONSE 14: We agree that there are situations where source and destinations nodes are known and as such should be specified in a connectivity analysis. For example, if the objective is to identify connections between a specific large wetland, which could be a source of frogs, and nearby ponds that could be potential destinations. But we were attempting to predict areas important for connectivity for multiple species throughout Canada. There are very few species for which all the sources and destinations are known, never mind multiple species. In addition, many species have continuous or semi-continuous populations in Canada, where sources and destinations lack meaning, whereas others species may have ranges that are spatially and/or temporally dynamic. We are also interested in identifying potential areas of connectivity for species that may not, in fact, be present at the moment. This can be important for planning for future connectivity (e.g., due to environmental change) or in anticipation of species recovery. We consider that the omnidirectional method was the appropriate method given our objectives, as others have in many other peer-reviewed studies. For example:

o Hohbein RR, Nibbelink NP. Omnidirectional connectivity for the Andean bear (Tremarctos ornatus) across the Colombian Andes. Landscape Ecology. 2021 Nov;36(11):3169-85.

o Choe H, Thorne JH. Omnidirectional connectivity of urban open spaces provides context for local government redevelopment plans. Landscape and Ecological Engineering. 2019 Jul;15(3):245-51.

o Gray ME, Dickson BG, Nussear KE, Esque TC, Chang T. A range‐wide model of contemporary, omnidirectional connectivity for the threatened Mojave desert tortoise. Ecosphere. 2019 Sep;10(9):e02847.

o McClure ML, Dickson BG, Nicholson KL. Modeling connectivity to identify current and future anthropogenic barriers to movement of large carnivores: a case study in the American Southwest. Ecology and evolution. 2017 Jun;7(11):3762-72.

o Fleishman E, Anderson J, Dickson BG. Single-species and multiple-species connectivity models for large mammals on the Navajo nation. Western North American Naturalist. 2017 Jul;77(2):237-51.

o Borja-Martínez G, Tapia-Flores D, Shafer A, Vázquez-Domínguez E. Highland forest’s environmental complexity drives landscape genomics and connectivity of the rodent Peromyscus melanotis. Landscape Ecology. 2022 Mar 15:1-9.

—-----------------------------------------

From your response: "current density patterns are biased by the placement of nodes (i.e., the same landscape will produce widely differing outputs depending on node placement)". As they should - if they weren't biased to node placement (reflecting our assumptions of where animals are), there would be a problem with our model. In that, I do not understand what movement, not dependent on node placement, means or ecologically signifies (because I cannot find the analogue for it in 'true' ecological patterns). Therefore, I do not know what the model will produce, and I do not know what to equate the model output with in terms of ecological processes. The authors need to clarify this, but have not.

RESPONSE 15: The model produces an estimate of current density, which is proportional to the probability of a random walk between all possible pairs of nodes, given the set of movement costs. We have added text in the methods to make certain this is clear (lines 182 - 183). Please see also Response 14. 

—---------------------------

In response 28, the authors say: "a detailed exploration of the theoretical underpinnings of the method are beyond the scope of our current paper" and I agree, but the authors will need to at least be clear on, and state, what their results mean ecologically.

RESPONSE 16: Our model provides a seamless estimate of the probability of movement during a random walk for a set of terrestrial species that use natural cover across Canada. Further research and analysis will allow us to continue to define the set of species and conditions that are well depicted by the model, but for now we have validated the model with several species in different regions of the country. We have made sure this is clearly described in the introduction at lines 137 – 146, and in the methods at lines 180 – 183 and 267 – 269. Also please see responses 14 and 15. 

—-------------------------------

I'd also point out that omnidirectional connectivity does not remove assumptions on node placement; they just move the nodes. Ie., you are assuming that animals want to move from the buffer of the map (where nodes are placed).

RESPONSE 17: Yes, we agree and this is exactly the point of the method. Nodes are moved to the buffer so that the resulting current density estimates inside of the modelled area of interest does not depend on node placement. 

Reviewer #3: Good morning everyone,

I want to warmly thank the reviewers for their open-mindedness relative to the comments we all raised after reading the original version of their manuscript. I (Reviewer #3 on the first round) am satisfied with most of the changes made and the way the manuscript has been revisited. This is clearer now, more straightforward, less balancing between an ecological story and a methodological paper; I really think that the manuscript is stronger this way.

Nevertheless, I still have some minor concerns that may deserve some attention before going forward in the publication process. I have listed them below, hoping that they could be interesting for the authors in the finalization of their revised manuscript, and for the Editorial board to take a decision regarding this promising paper.

1) General comment #1 – structure (mainly discussion): Excellent revision of the structure of your manuscript, congratulations! The flow of arguments is clearer now, thank you. Nevertheless, I suggest adding another subtitle in the discussion, ~line 487, as the paragraph here (lines 487-511, maybe lines 487-529) seem to belong to another “section” compared to lines 432-485). While the lines 487-511 and 524-529 refer to the location of the corridors, lines 513-522 could be moved to the limitations or even more interestingly to the methods. In contrast, lines 433-485 explain more in details the strength of the exercise and the validation process.

RESPONSE 18: Thank you very much for taking the time to provide such helpful suggestions, which we feel have made the manuscript much better. We have inserted a section break as suggested (line 658).

Note: all line numbers correspond to those in “Revised_Manuscript_with_Track_Changes.doc” when viewed using ‘All Markup’ in Track Changes.

—----------------------------------------

2) Methods (lines 260-262): I suggest adding a few references in support to this decision, as the 10% of the GPS data sample for each individual seems quite a random decision and a random threshold.

Response 19. Thank you for this comment. We added to lines 342 - 344 that we randomly selected 10% of the data “as this is a common testing ratio used in habitat model validations (e.g., Heinemeyer et al. 2019) and can reduce the effects of autocorrelation”. See figures below for comparisons of model parameters estimated using different testing ratios). We tested three different testing ratios (100%, 10%, and 5%) and they show similar mean coefficient values and statistical significance across them. 

Comparison of coefficient estimates for the species-specific linear mixed effects models (without interaction term) fitted to three different testing ratios of the GPS telemetry data: 100%, 10% and 5% of each individual’s data. From top left to bottom right, panels depict coefficient estimates for grey wolf, mountain caribou, moose and Rocky Mountain elk, Coefficient values for “Observed” depict the mean difference in current density values between observed GPS locations and available, but unused locations. 

Comparison of coefficient estimates for the species-specific linear mixed effects models (with interaction term) fitted to three different testing ratios of the GPS telemetry data: 100%, 10% and 5% of each individual’s data. From top left to bottom right, panels depict coefficient estimates for grey wolf, mountain caribou, moose and Rocky Mountain elk.

—----------------------------------------

3) Methods (lines 294-296 and 303-304): Again, please provide some support to the decision of using a 300-m radius buffer for the herpetofauna roadkill data, and a 900 x 900m windows along roads for the moose roadkill data. In a recent paper, we (Laliberté & St-Laurent 2020, Landscape and Urban Planning) have shown that the radius buffer could influence (slightly, but this is a real effect) the strength of the correlation during the validation process. I suspect that maybe testing different buffer sizes would have influenced, even very slightly, the validation strength.

RESPONSE 20: We have added text to line 408 to clarify that used a 300 m radius around point data for herpetofauna roadkill to identify which pixels to use when taking the mean current density. Whereas because moose roadkill was so abundant and widespread, we ‘pre’ divided roads into 900 m sections (text updated in lines 414 – 416). 

We also now note in the potential limitations section of the Discussion (lines 748 - 753) the role of buffer size. We are interested in further exploring the role of scale when using road kill data to validate connectivity in a future project. For this study, however, we were interested in the local scale; i.e., connections proximate to the road. 

—----------------------------------------

4) Results (lines 316-318): I am not convinced that this reason (“… because that value has been used in previous studies”) is sufficient (or appropriate). CircuitScape cannot deal with null values, that’s why we usually add “+1” to all resistance values.

RESPONSE 21: Thank you. The manuscript has been revised to reflect your point (line 430).

—---------------------------------------

5) Results (line399-401 and Fig. 7): I think this could be important to bring to the attention of the authors that even if a positive, significant correlation has been found between current density and moose roadkill, this relationship is quite weak (see the very low Pearson r on Fig. 7). Looking at the figure suggest that this relationship is very “noisy” with a lot of variation in the number of roadkills for the same current density. This would open the opportunity to discuss the reliability of the different validation metrics in your discussion (as we did in Laliberté & St-Laurent 2020), and to identify potential other factors that may mask this expected relationship.

RESPONSE 22: We have revised the Results to be clear it was a weak correlation for moose in New Brunswick (line 517) and it was already mentioned in the Discussion (line 641 - 643). We have also added some discussion regarding other potential factors influencing roadkill hotspots, with examples (lines 645 - 651) and contrast that data type with GPS data as a possible explanation for why different types of data performed differently for validation (lines 651 - 655)

—---------------------------------------

6) Discussion (lines 462-463): Along with my comment just above, here you have to spend a little bit more time explaining why the different validation metrics performed differently, especially the moose roadkills. I suggest to view your result critically even if the relationship follows what was expected.

RESPONSE 23. Please see response 22.

—---------------------------------------

7) Discussion (lines 483-485): please add some examples, supported by the literature, for these “other factors”.

RESPONSE 24. Added (Lines 645 - 651).

—---------------------------------------

8) Discussion (Limitations): I REALLY enjoyed your section about potential limitations, nice job. You might have to consider talk about the “vignetage” issue that may be at play in some maps (see for examples Fig. 4a-b-c), but this issue needs more explanation in the methods (that’s why you overlapped some tiles…), especially regarding official boundaries (e.g. USA vs. Canada).

RESPONSE 25. We have revised the text to better explain how we used tiles and buffers to address the effects of node placement on current density in the Methods section, lines 254 - 255 and 261 - 267 and updated the map to better display the tiles in S4 Fig. We also added some text in the Methodological Advances section of the Discussion that discusses the issue of the anomalies in two regions (lines 719 - 726).

---

## [Editor Report · Decision Letter 2]

7 Feb 2023

Predicting Areas Important for Ecological Connectivity Throughout Canada

PONE-D-22-17312R2

Dear Dr. Pither,

We’re pleased to inform you that your manuscript has been judged scientifically suitable for publication and will be formally accepted for publication once it meets all outstanding technical requirements.

Kind regards,

Julian Aherne

Academic Editor

PLOS ONE

Additional Editor Comments (optional):

I commend the authors for their thoughtful and thorough responses to the reviewers’ comments; the revised manuscript addresses all reviewers concerns and is acceptable for publication. Well done.
---

## [Editor Report · Acceptance letter]

13 Feb 2023

PONE-D-22-17312R2 

Predicting Areas Important for Ecological Connectivity Throughout Canada 

Dear Dr. Pither:

I'm pleased to inform you that your manuscript has been deemed suitable for publication in PLOS ONE. Congratulations! Your manuscript is now with our production department. 

Kind regards, 

on behalf of

Dr. Julian Aherne 

Academic Editor

PLOS ONE